# Local moisture recycling across the globe

Jolanda J.E. Theeuwen[1,2], Arie Staal[1], Obbe A. Tuinenburg[1], Bert V.M. Hamelers[2,3], Stefan C. Dekker[1]

[1]Copernicus Institute of Sustainable Development, Utrecht University, Utrecht, 3584 CB, The Netherlands
[2]Wetsus, European Centre of Excellence for Sustainable Water Technology, Leeuwarden, 8911 MA, The Netherlands
[3]Department of Environmental Technology, Wageningen University and Research, Wageningen, 6708 PB, The Netherlands

*Correspondence to*: Jolanda J.E. Theeuwen (j.j.e.theeuwen@uu.nl)

**Abstract.** Changes in evaporation over land affect terrestrial precipitation via atmospheric moisture recycling and consequently freshwater availability. Although global moisture recycling at regional and continental scales are relatively well understood, the patterns of local moisture recycling and the main variables that impact it remain unknown. We calculate the local moisture recycling ratio (LMR) as the fraction of evaporated moisture that precipitates within a distance of 0.5° (typically 50 km) from its source, identify variables that correlate with it over land globally and study its model dependency. We derive seasonal and annual LMR using a 10-year climatology (2008–2017) of monthly averaged atmospheric moisture connections at a scale of 0.5° obtained from a Lagrangian atmospheric moisture tracking model. We find that, annually, on average 1.7% (st.dev. = 1.1%) of evaporated moisture returns as precipitation locally, but with large temporal and spatial variability, where LMR peaks in summer and over wet and mountainous regions. Our results show that wetness, orography, latitude, convective available potential energy, wind speed, and total cloud cover correlate clearly with LMR, indicating that especially wet regions with little wind and strong ascending air are favourable for high LMR. Finally, we find that spatial patterns of local recycling are consistent between different models, yet the magnitude of recycling varies. Our results can be used to study impacts of evaporation changes on local precipitation, with implications for, for example, regreening and water management.

## 1 Introduction

Atmospheric moisture connections redistribute water from evaporation sources to precipitation sinks, affecting climates globally, regionally, and locally. These connections are key in the global hydrological cycle and are used to understand the importance of terrestrial evaporation for water availability. As evaporated moisture can travel up to thousands of kilometres in the atmosphere, changes in evaporation can affect precipitation in a large area. An evaporationshed (Van der Ent and Savenije, 2013) describes where evaporated moisture from a specific source region precipitates and therefore, can be used to study (1) the changes in precipitation on a global scale following a change in evaporation in the source region and (2) atmospheric moisture recycling. Globally, more than half of terrestrial evaporated moisture precipitates over land (Van der Ent et al., 2010; Tuinenburg et al., 2020), which is a process called terrestrial moisture recycling. About half of terrestrial precipitation originates from land (Tuinenburg et al., 2020). Hence, terrestrial moisture recycling has an important contribution to water availability. For example, 80% of China's water resources originates from evaporation over Eurasia (Van der Ent et

al., 2010). Furthermore, areas can also feed precipitation to themselves through regional moisture recycling. In the Amazon
basin, 63% of the evaporated moisture precipitates within the basin itself (Tuinenburg et al., 2020). Terrestrial moisture
recycling is considered an ecosystem service (Falkenmark et al., 2019; Keys et al., 2016) as globally, almost 20% of terrestrial
precipitation originates from vegetation-regulated moisture recycling (Keys et al., 2016). How this ecosystem service is
affected by, for instance, deforestation, can be studied using atmospheric moisture connections.

Moisture recycling has been used to study downwind impacts of land-use changes (e.g. Bagley et al., 2012; Keys et al., 2012;
Wang-Erlandsson et al., 2018), which can affect both the magnitude and pattern of moisture recycling (Van der Ent et al.,
2014), and the impact of ecosystems on other ecosystems (e.g. O'Connor et al., 2021). Hence, atmospheric moisture
connections can be used for freshwater governance to understand and manage the impacts of land-use changes downwind such
as changes in freshwater availability for irrigation and plants. (te Wierik et al, 2021; Te Wierik et al, 2020). For example,
previous research showed that for 45% of the land surface, an increase in vegetation is beneficial for downwind water
availability (Cui et al., 2022).

So far, analytical recycling models and moisture tracking models have been used to study terrestrial recycling and downwind
impacts of land cover change on global and regional levels (Burde & Zangvil, 2001; Van der Ent et al., 2010). Multiple studies
focus on the regional recycling for specific regions, with a spatial scale ranging from 500 km up to several thousands of
kilometres (e.g., Burde, 2006; Dominguez et al., 2006; Lettau et al, 1979; Staal et al., 2018; Trenberth, 1999). Furthermore,
regional recycling on a spatial scale of 1.5° has been studied globally using a Eulerian moisture tracking model, assuming a
well-mixed atmosphere (Van der Ent and Savenije, 2011). It was debated that regional recycling ratios are difficult to compare
due to differences in the shape and size of the studied regions (Van der Ent and Savenije, 2011). Therefore, Van der Ent &
Savenije (2011) defined the typical length scale of evaporation recycling, which can be used to compare between different
regions because it is independent of the size and shape of a regions. This length scale decreases with increasing regional
recycling and, therefore, is a proxy for an area's regional recycling. However, it does not allow for the quantification of the
amount of water that recycles within the defined region and therefore does not provide quantitative insight into the regional
impacts of evaporation changes induced by land-cover changes.

In regions with a high regional recycling, reforestation can enhance freshwater availability and for regions with a low recycling,
reforestation may cause local drying (Hoek van Dijke et al., 2022) due to reductions in streamflow as a result of enhanced
evaporation locally (Brown et al., 2005; Jackson et al., 2005). To physically understand, for instance, the role of local wetting
or drying due to reforestation, deforestation, or the use of groundwater or surface water for irrigation, local moisture recycling
is key. We argue that local impacts need to be studied explicitly as they may have a crucial role in future water governance,
e.g., to prevent tree restoration projects causing local drying.

The state-of-the-art high-resolution atmospheric moisture connections obtained with the Lagrangian atmospheric moisture
tracking model "UTrack" allows us to calculate the evaporation recycling ratio at higher spatial resolution (0.5°) (Tuinenburg
et al., 2020; Tuinenburg and Staal, 2020). We define this as the local moisture recycling ratio (LMR) as this high resolution
allows us to study local-scale land-atmosphere feedbacks, which will help us better understand hydrological impacts of land-
use change. LMR describes which fraction of evaporated moisture recycles within its source grid cell and its eight surrounding
grid cells. Moisture recycling has not been studied before on this high-resolution scale globally. To get a better physical
understanding of this metric we identify which factors correlate with it. We analyse this for different latitude classes to account
for different cell sizes across latitude. Factors included in this analysis are: orography, precipitation, precipitation type,
evaporation, shear, convective available potential energy, atmospheric moisture flux, wind speed, total cloud cover, boundary
layer height and surface net solar radiation. These variables relate to either convection, local wetness, or moisture transport
away from the source location, which we identified as important factors for local moisture recycling. Furthermore, we study
how LMR varies over the globe and throughout the year for a 10-year climatology (2008-2017), as well as its scaling and
model dependency.
**2 Methods**
We use global atmospheric moisture connections obtained from Tuinenburg et al., (2020) to calculate LMR worldwide. These
moisture connections are a 10-year climatology (2008–2017) of monthly averages and have a spatial resolution of 0.5°. These
UTrack-atmospheric-moisture data are derived using a Lagrangian atmospheric moisture tracking model by Tuinenburg &
Staal (2020) that tracks evaporated moisture at a spatial scale of 0.25°, and stored at a spatial resolution of 0.5°. In this model,
for each grid cell of 0.25°, each mm of evaporation is represented by one hundred released moisture parcels. The wind
transports these parcels horizontally and vertically through the atmosphere. Additionally, a probabilistic scheme describes the
vertical movement of the moisture parcels over 25 atmospheric layers. In this scheme, the parcels are randomly distributed
across the vertical moisture profile of each grid cell. At each time step (0.1 h), the moisture budget is made using evaporation,
precipitation and total precipitable water. Parcels are tracked for up to 30 days or up to the point at which only 1% of their
original moisture is still present. On average, the lifetime of atmospheric moisture is 8-10 days (Sodemann, 2020). However,
some moisture might still remain in the parcels after 10 days. After 30 days for most of the parcels all of the original moisture
has rained out (Tuinenburg and Staal, 2020). Input data for UTrack consist of evaporation, precipitation, precipitable water,
and wind speed obtained from the ERA5 dataset (Hersbach et al., 2020). We refer to Tuinenburg & Staal (2020) for a complete
description of the model settings and the tests and assumptions underlying them.

LMR is the fraction of evaporated moisture that precipitates locally. To study the scale dependency of local moisture recycling,
we examine three definitions of LMR (Fig. A1): the fraction of evaporated moisture that precipitates in f(1) its source grid
cell, i.e., $r_1$, (2) its source grid cell and its eight neighbouring grid cells, i.e., $r_9$, and (3) its source grid cell and its 24
neighbouring grid cells, i.e., $r_{25}$. Equations 1-3 describe the three definitions of LMR, in which $E_{i,j}$ is the amount of moisture
evaporated from source grid cell $i,j$. The fraction of $E_{i,j}$ that precipitates within its source grid cell and its (8 or 24) neighbouring
grid is indicated by $P_{E,i+l,j+k}$ ($i+l,j+k$, with $l = 0$ and $k = 0$ for $r_1$, $l = -1,0,1$ and $k = -1,0,1$ for $r_9$ and $l = -2,-1,0,1,2$ and $k = -2,-$
$1,0,1,2$ for $r_{25}$ ).
$r_1 = \dfrac{P_{E,i,j}}{E_{i,j}}$

102          (1)

$r_9 = \dfrac{\sum_{l=-1}^{1} \sum_{k=-1}^{1} P_{E,i+l,j+k}}{E_{i,j}}$            (2)
$r_{25} = \dfrac{\sum_{l=-2}^{2} \sum_{k=-2}^{2} P_{E,i+l,j+k}}{E_{i,j}}$            (3)

$r_1$, $r_9$, and $r_{25}$ result in different local moisture recycling ratios across the globe (Fig. A2). $r_1$ peaks over the ocean where
precipitation is relatively low and evaporation is relatively large, which results in relatively large recycling ratios. In addition,
we find exceptionally low values over mountain peaks, yet not over all elevated terrain. This result is inconsistent with the
patterns found for $r_9$ and $r_{25}$, as these patterns include peaks over mountainous and low recycling over the oceans. These
patterns can be explained by enhanced convection over mountains due to orographic lift and strong winds over the ocean that
carry moisture away from its source. The patterns found for $r_9$ and $r_{25}$ seem to capture multiple physical processes that are
important for moisture transport and formation of precipitation better than the pattern of $r_1$. In our study we do not focus on $r_1$,
as $r_1$ does not include all small-scale flows of <50 km. This is because moisture can evaporate from cell i,j, and precipitate in
the adjacent cell, while transport length is <50 km. Furthermore, as the patterns of $r_9$ and $r_{25}$ are similar and agree with our
understanding of relevant processes, we decided to define the local moisture recycling ratio (LMR) as $r_9$ to keep the spatial
scale as small as possible. For $r_9$, the distance between the center of the source grid cell and its surrounding grid cells describes
the typical length of the local moisture flow, which is approximately 0.5°. We calculated this typical length across the globe
by calculating the average of the average zonal length, meridional length, and diagonal length of all terrestrial grid cells. The
total average equals 50.1 km (st.dev. = 15.5 km), so, the average moisture flow length is approximately 50 km.

Furthermore, the LMR derived with the Lagrangian approach using output from UTrack is compared with the output from the
Eulerian moisture tracking model WAM2-layers (Link et al., 2020), to study the model dependency of LMR. For this
comparison, the resolution of the UTrack data is reduced to 1.5° to match the output of the WAM2-layers model. To do so, all
evaporationsheds over land were multiplied with their source evaporation. Then, the recycling within cells of 1.5° was
calculated for all terrestrial surfaces. A detailed description of the atmospheric moisture connections obtained with WAM2-
layers and the model itself are provided by Link et al. (2020) and Van der Ent et al. (2013).

We study the relations between multiple variables and the 10-year climatology (2008-2017) of local moisture recycling to
identify factors that relate to recycling to assess what factors might affect recycling. To calculate this 10-year climatology of
LMR, for each month, we weighted the multi-year (2008-2017) monthly LMR by multi-year monthly evaporation in the same
period:
$LMR_{annual\ average} = \sum_{i=jan}^{dec} LMR \frac{E_{month\ i}}{E_{year}}$ (4)
in which $E_{year}$ is the sum of the evaporation of the 12 months. To identify factors that might affect LMR, variables that relate
to atmospheric moisture and vertical displacement of air, as both higher atmospheric moisture content and ascending air
promote precipitation are selected. All these variables are obtained, either directly or indirectly from ERA5 reanalysis data
(Hersbach et al., 2020). We downscaled the original resolution from 0.25° to 0.5° by centrally averaging the data. For all
variables we calculated the climatological mean for the years 2008-2017.

In total 13 variables are selected (Fig. A3): (1) elevation ($z$) which we expect to enhance LMR through orographic lift. (2)
Precipitation, which we expect to correlate positively with LMR given that in Lagrangian moisture tracking models, the amount
of moisture that leaves the parcel (i.e., precipitates) scales with precipitation. (3) Total evaporation as it enhances the
atmospheric moisture content and we, therefore, expect it to promote precipitation locally. (4) Wetness (precipitation minus
evaporation), as with increasing wetness the downward flux of moisture increases and evaporated water becomes more likely
to precipitate, possibly promoting LMR. (4) Convective precipitation and (5) large-scale precipitation, as they scale with
precipitation, by definition. Both are included to study whether the type of precipitation is an important factor explaining LMR.
(6) Latitude, which is a proxy for processes related to the Hadley cell circulation, which is characterized by strong ascent and
descent of air at specific latitudes, which we expect to have an important contribution to LMR, because they respectively
enhance and reduce the formation of precipitation (Wang and Yang, 2022). (7) The vertical integral of the atmospheric
moisture flux (in northward and eastward directions and the total flux) as it carries the moisture away from its source and could
thus reduce LMR. (8) Convective available potential energy (*CAPE*), which feeds convection and therefore promotes
precipitation locally, which could enhance LMR. (9) Vertical wind shear between 650 and 750 hPa of both meridional and
zonal winds, as it affects moisture transport in multiple directions and, therefore, we expect it to impact LMR. (10) Total wind
speed, as it carries the wind, and therefore, we expect it to correlate negatively with LMR. (11) Total cloud cover as a proxy
for condensation processes which possibly enhance LMR (Richards and Arkin, 1998). (12) Boundary layer height, because
thinner boundaries need less evaporation to reach saturation of air, and therefore, we expect it will promote precipitation
locally. Finally, (13) net surface solar radiation as a proxy for the energy source of convection, and other processes, which we
expect to be important for LMR. We calculate shear ($\tau$) using Equation (5).
$\tau = \frac{ln\frac{v_2}{v_1}}{ln\frac{z_2}{z_1}}$ (5)
In this equation, $v_1$ and $v_2$ are the wind speed (in zonal and meridional directions) at two different heights ($z_1$ and $z_2$). We
identified significant correlations using Spearman rank correlations. It should be noted that a correlation does not imply
causality. We exclude oceans, seas and Antarctica from this analysis using the land-sea mask from ERA5. We classify the data
based on latitude to account for decreasing grid cell size with increasing latitude. Each class has a range of 15° and includes
the grid cells on both the Northern and Southern Hemispheres (see Table A1). Between 60° and 90° south, the grid cells do
not contain land besides Antarctica, and are therefore not included in the classes. Additionally, we used the Ecoregions 2017
data (https://ecoregions.appspot.com/) to study the spatially averaged local moisture recycling of 14 biomes across the globe
(Fig. A4). We study variation amongst biomes, as biomes include information on both biotic factors such as vegetation type,
and abiotic factors such as climate.
**3 Results**
**3.1 LMR obtained from output of UTrack**
Annually, on average about 1.7% (st. dev. = 1.1%) of terrestrial evaporated moisture recycles locally. LMR shows spatio-
temporal variation (Fig. 1) with peaks over elevated (e.g., the Atlas Mountains and Ethiopian Highlands) and wet areas (e.g.,
Congo Basin and Southeast Asia) and minima over arid regions (e.g., Australia and the Sahara Desert). Additionally, we find
peaks in LMR during summer (i.e., during DJF for the Southern Hemisphere and during JJA for the Northern Hemisphere).
This seasonality is especially strong over mountainous and wet areas. For the mid-latitudes, especially the Mediterranean Basin
shows seasonality with peaks in summer (JJA). However, seasonality is largest at low latitudes. Within the tropics we find
some spatial differences. First, LMR in the Congo Basin and Southeast Asia exceed LMR in the Amazon Basin. Second,
recycling in the Congo Basin and Southeast Asia peaks in JJA and recycling in the Amazon Basin peaks in DJF, which is the
wet season for a large part of the Amazon.

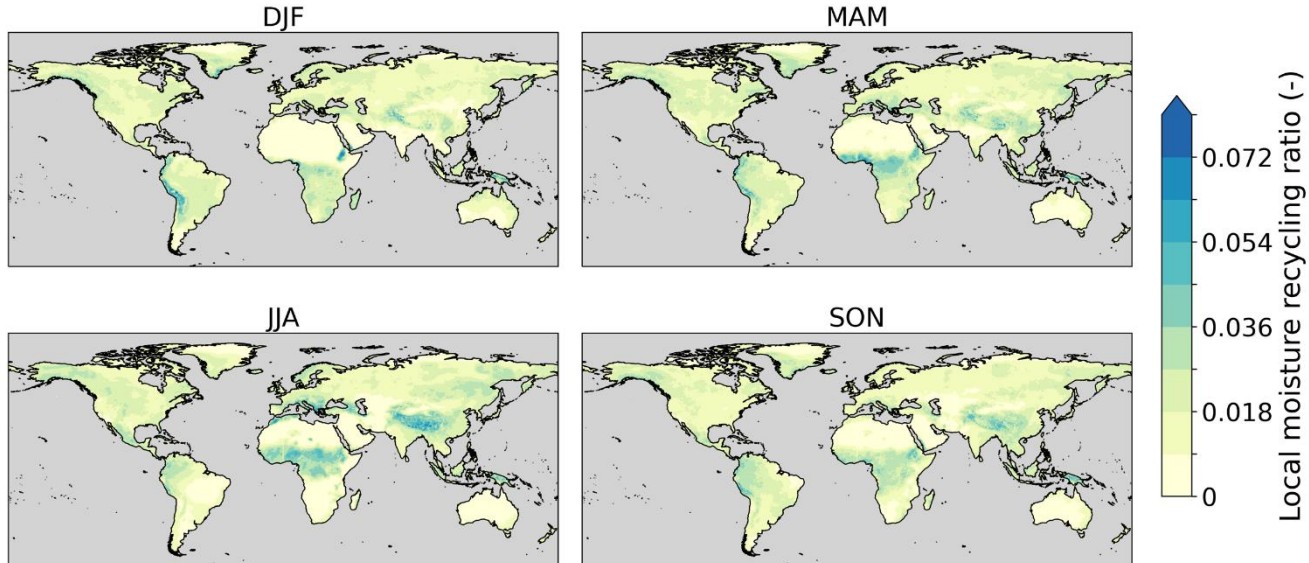


**Figure 1. 10-year climatology (2008–2017) of the seasonal averages of local moisture recycling across the global land surface. Here, local moisture recycling is defined as the fraction of evaporated moisture that precipitates in its source grid cell and its eight neighbouring grid cells (r9). Different seasons are DJF: December–February, MAM: March–May, JJA: June–August, and SON: September–November.**

We calculated recycling on a 1.5° grid using both the dataset by Link et al. (2020), which we refer to as $r_{WAM2\text{-}layers}$, and the dataset by Tuinenburg et al. (2020) (upscaled to 1.5°), which we refer to as $r_{UTrack}$, to study the model dependency of local recycling (Fig. 2). We find that the global spatial patterns of $r_{UTrack}$ and $r_{WAM2\text{-}layers}$ agree (Fig. 2 & Fig. A5). However, the magnitude of $r_{WAM2\_Layers}$ is larger than $r_{UTrack}$ over mountains, the tropics, and the high latitudes. $R_{Utrack}$ is larger than $r_{WAM2\text{-}layers}$ over drylands and deserts (e.g., the Sahel region and Western Asia) (Fig. 2). However, over drylands and deserts recycling ratios are relatively small and therefore, the relative difference as presented in Fig. 2 has less meaning here. Globally, the difference between $r_{UTrack}$ and $r_{WAM2\text{-}layers}$ and its variation is largest around the equator (Fig. A6). On average, the relative difference between UTrack and WAM2-Layers ((UTrack-WAM2-Layers)/ UTrack) equals -1.5 (st.dev. = 3.4).

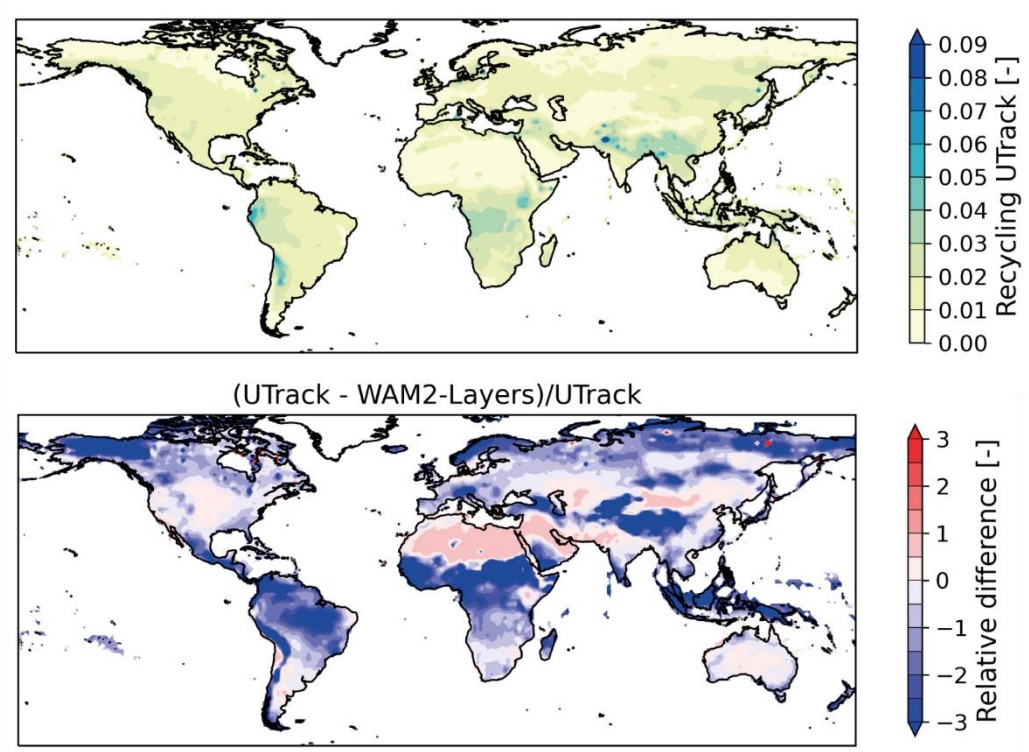

**Figure 2.** $r_{Utrack}$ , the fraction of evaporated moisture that recycles within its source grid cell of 1.5° (top). The relative deviation between $r_{Utrack}$ and $r_{WAM2-layers.}$(bottom). This deviation is calculated using the recycling within one grid cell at a resolution of 1.5° obtained from the datasets of Tuinenburg et al. (2020) and Link et al. (2020).

## 3.2 Factors underlying LMR

For each latitude class we calculated the Spearman rank correlation coefficient (ρ) (Table 1). Below we discuss only statistically significant (p<0.05) correlations with ρ ≥ 0.4, that indicate a moderate correlation. These correlations are emboldened in Table 1. We find that LMR correlates positively with total precipitation and wetness for all classes between 15° and 75°. In addition, between 15° and 30°, LMR correlates strongly with total precipitation (ρ = 0.80). Besides total precipitation, large scale precipitation (between 15° and 45° and between 60° and 75°) and convective precipitation (between 15° and 45°) correlate positively with LMR. Overall, there is a moderate correlation between precipitation and LMR for the mid-latitudes. The highest correlation between LMR and convective precipitation is found between 15° and 30° latitude. For this latitude class, LMR also correlates positively with evaporation and CAPE. Despite the low correlation between LMR and CAPE, for most of the latitude classes, high CAPE clearly relates to LMR, as the skewed profile in the scatter density plot indicates that only a small amount of the grid cells with a relatively high CAPE have a low LMR (Fig 3). Furthermore, the presence of clouds also correlates with LMR. Between LMR and total cloud cover, a positive correlation holds between 15° and 45°, and a negative correlation holds between 60° and 75°. Furthermore, the vertical integrals of the eastward and

northward moisture fluxes correlate less with LMR compared to the vertical fluxes (e.g., precipitation). For the higher latitudes,
the northward moisture flux correlates positively with LMR (between 60° and 75°) and the eastward moisture flux correlates
negatively with LMR (between 75° and 90°). The moisture flux depends on wind speed, yet, wind speed correlates negatively
with LMR for the lower latitudes (between 0° and 45°). Furthermore, LMR correlates positively with orography between 30°
and 75°. We find that for high elevation, LMR is always relatively high (Fig A7). Additionally, LMR correlates negatively
with boundary layer height between 45° and 60°. Finally, LMR correlates negatively with wind shear at 650 hpa in the
meridional direction (between 75° and 90°) and with latitude (between 60° and 75°). However, we find an oscillating relation
between LMR and latitude (Fig 4), which is not captured by the Spearman rank correlation coefficients. This pattern indicates
high LMR over the equator (0°) and 60° north, and low LMR around 30° north and south. Orography seems to disrupt the
relation between latitude and LMR causing peaks in LMR around 35° north and 20° south (Fig 4). LMR does not correlate to
surface net solar radiation for any latitude. However, for low surface net solar radiation (<0.75*$10^6$ J/m²) holds that LMR
increases with increasing surface net solar radiation (Fig 3).

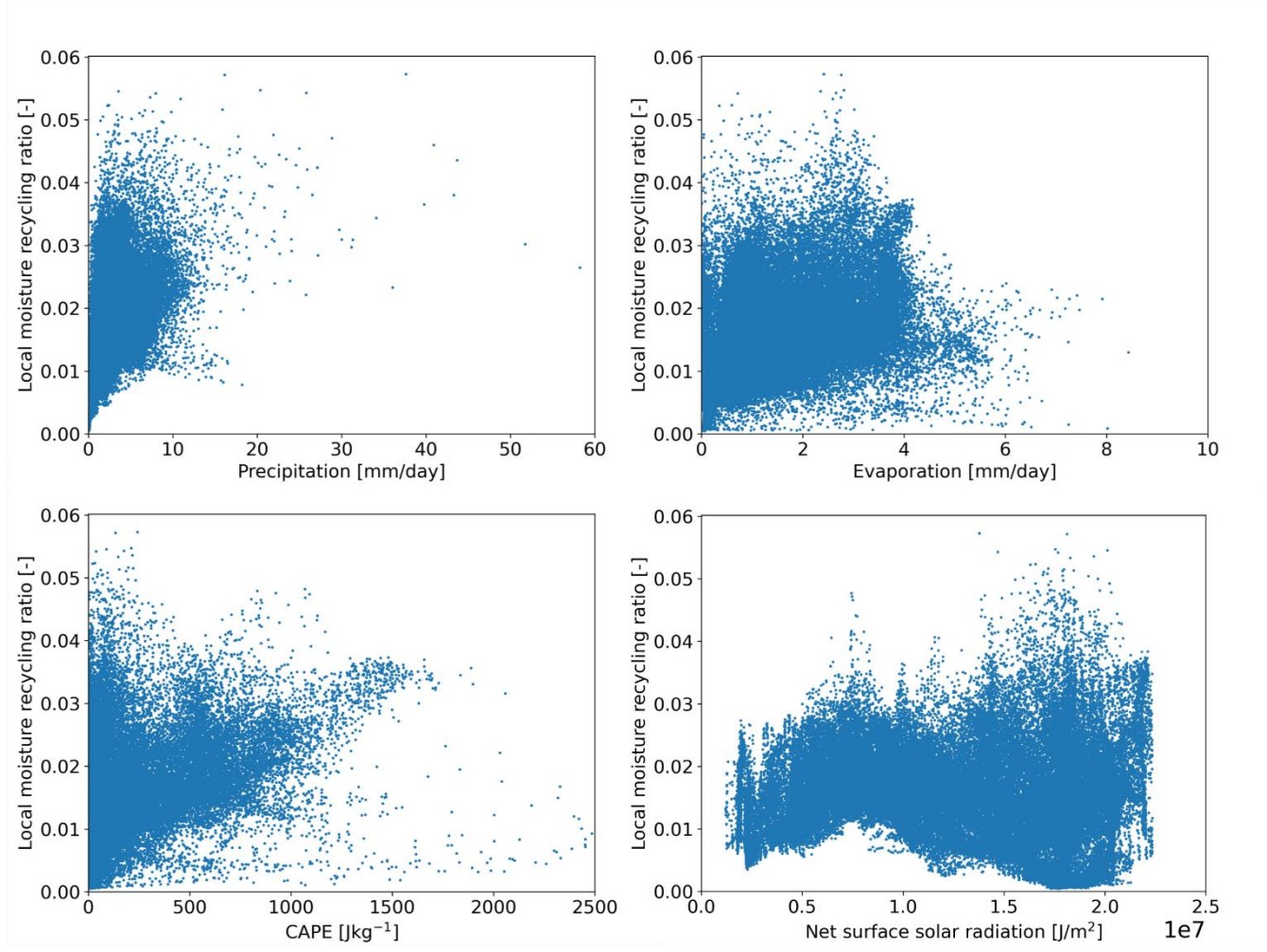


**Figure 3: Scatter plots of the 10-year climatology (2008–2017) of the local moisture recycling ratio over land and precipitation (top**
**left), evaporation (top right), convective available potential energy (CAPE) (bottom left), and solar net surface radiation (bottom**
**right). Each dot represents a 0.5° resolution grid cell over land.**

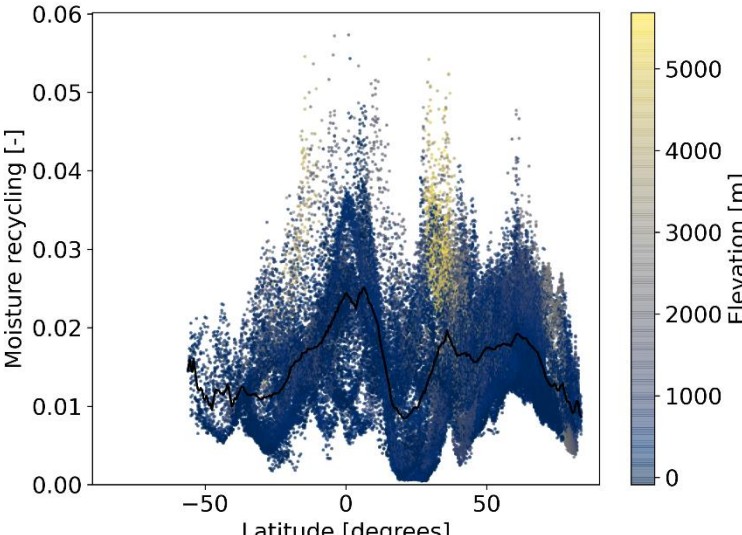


**Figure 4. Scatter plot of the 10-year climatology (2008-2017) of LMR and latitude. The colour scale indicates elevation, with blue**
**being low elevation and yellow being high elevation. The black line represents the zonal average of LMR. Each dot represents a**
**0.5° resolution grid cell over land.**

Table 1. Spearman rank correlation coefficients between LMR and all tested variables. '*' indicates a significant correlation (p<0.05) and moderate and strong relations (ρ>0.4) are emboldened. The classes including latitudes between 0° and 60° include grid cells of the Northern Hemisphere and Southern Hemisphere. The classes including latitudes exceeding 60° include grid cells of the Northern Hemisphere only.

| | Spearman rank correlation coefficient | | | | | |
|---|---|---|---|---|---|---|
| **Variable** | **0°-15°** | **15°-30°** | **30°-45°** | **45°-60°** | **60°-75°** | **75°-90°** |
| Total precipitation (P) | 0.15* | **0.80*** | **0.47*** | **0.40*** | **0.45*** | 0.37* |
| Total evaporation (E)_ | -0.05* | **0.63*** | 0.19* | -0.12* | 0.19* | 0.20* |
| Wetness (P-E) | 0.18* | **0.59*** | **0.52*** | **0.48*** | **0.43*** | 0.27* |
| Convective precipitation | 0.20* | **0.79*** | **0.46*** | 0.29* | 0.35* | 0.33* |
| Large scale precipitation | -0.06* | **0.75*** | **0.46*** | 0.38* | **0.40*** | 0.36* |
| Fraction of cp | 0.36* | -0.35* | -0.13* | -0.14* | 0.19* | 0.28* |
| Latitude | 0.24* | -0.18* | 0.22* | 0.14* | **-0.40*** | -0.18* |
| Eastward moisture flux | 0.15* | 0.00 | -0.30* | -0.38* | -0.20* | **-0.49*** |
| Northward moisture flux | -0.03* | 0.22* | 0.29* | -0.03* | **0.48*** | 0.23* |
| Total moisture flux | -0.28* | 0.30* | -0.29* | -0.33* | -0.03 | -0.16* |
| CAPE | 0.31* | **0.58*** | 0.37* | 0.06* | 0.12* | -0.02 |
| Zonal shear | 0.15* | -0.12* | 0.02 | -0.31* | 0.00 | 0.24* |
| Meridional shear | -0.22* | 0.15* | -0.08* | -0.01 | 0.05* | **-0.46*** |
| Orography | 0.31* | 0.29* | **0.49*** | **0.54*** | **0.68*** | -0.13* |
| Total cloud cover | 0.28* | **0.78*** | **0.43*** | 0.09* | **-0.56*** | 0.08* |
| Surface net solar radiation | -0.16* | 0.10* | -0.30* | -0.08* | 0.28* | 0.21* |
| Boundary layer height | -0.31* | -0.32* | -0.39* | **-0.53*** | -0.18* | -0.06* |
| Total wind speed | **-0.46*** | **-0.55*** | **-0.47*** | -0.26* | -0.26* | -0.30* |

## 4 Discussion

### 4.1 Factors underlying LMR

Moisture recycling affects humanity by influencing water security, agriculture, forestry, regional climate stability and Earth system resilience (Keys et al., 2019; Wang-Erlandsson et al., 2022). Different types of moisture recycling were subject to research used for different applications (e.g., Bagley et al., 2012; Pranindita et al., 2022; Van der Ent et al., 2010). We analysed the local moisture recycling ratio (LMR) (of evaporated moisture) across the globe at 0.5° resolution, and which factors might affect it. We find that LMR, defined as the fraction of evaporated moisture that precipitates within a distance of 0.5° (typically

50 km) from its source, varies over time and space, peaking in summer and over elevated and wet regions. First, latitude,
elevation, and Convective Available Potential Energy (CAPE) seem to be important factors influencing LMR (Fig. 5). These
variables all promote convection (Roe, 2005; Scheff and Frierson, 2012; Wallace and Hobbs, 2006), strongly suggesting a
dependency of LMR on convection. Convective storms develop due to unstable conditions resulting in precipitation locally
(Eltahir, 1998) and a higher CAPE results in more rainfall (Eltahir and Pal, 1996; Williams and Renno, 1993). The pattern of
LMR across latitudes also coincides with updraft and downdraft of air caused by the Hadley cell circulation (Wallace and
Hobbs, 2006). Around the equator and 60° north and south, air ascends, where we find a high LMR. Around 30° north and
south, air descends, where we find a low LMR. Deviations from this pattern correspond to higher elevations which promote
LMR through orographic lift. Overall, our results suggest a positive relation between convection and LMR.

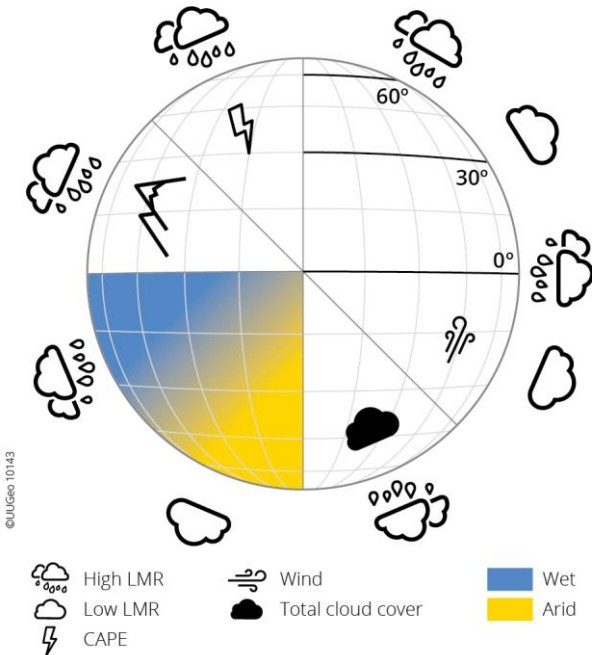


**Figure 5. Conceptual model of the most important factors influencing local moisture recycling around the globe. Rainy clouds indicate variables that increase LMR and clouds without raindrops indicate variables that decrease LMR. Blue indicates wet regions, yellow indicates arid regions.**

Second, wetness seems to be an important factor underlying LMR as LMR significantly correlates with precipitation and *P-E*
(precipitation minus evaporation). Furthermore, both large-scale and convective precipitation significantly correlate with
LMR. This is surprising, as convection promotes precipitation locally (Eltahir, 1998); therefore, we expected a stronger
correlation between LMR and convective precipitation than between LMR and large-scale precipitation. As both correlations
are similar, this suggests that the type of precipitation might not affect LMR. Although convection is a local-scale process (i.e.,
having a spatial scale below 100 km) (Miyamoto et al., 2013), remotely evaporated moisture can be transported to a region
with high convective activity and then precipitate as convective precipitation (Jana et al., 2018; Liberato et al., 2012). In that

way, the precipitation type is independent of the distance between moisture source and target location and therefore does not relate to LMR. Total cloud cover correlates both positively (between 15° and 45°) and negatively (between 60° and 75°) with LMR. Total cloud cover correlates with precipitation, convective precipitation, and large-scale precipitation for all latitudes except between 60° and 75° (Tab. A2). Due to the positive correlation between LMR and precipitation and the absence of a correlation between precipitation and total cloud cover at these latitudes we can statistically explain the negative correlation between total cloud cover and LMR. Physically, this result is harder to explain. Our results describe the importance of convection underlying LMR at lower latitudes, where total cloud cover correlates with convective precipitation. For higher latitudes, the importance of convection underlying LMR decreases, and we therefore expected also the correlation between total cloud cover and LMR to decrease but not to become negative. Likely, another process that we cannot identify with our analysis causes the correlation between total cloud cover and LMR to be negative. Overall, we find that wetness enhances LMR independent of the precipitation type.

Unexpectedly, we do not find a clear correlation between the vertical integral of the atmospheric moisture flux and LMR. However, for the lower latitudes (between 0° and 45° latitude), LMR correlates to wind speed (at 10 and 100 m) which carries evaporated moisture away from its source location, enhancing the moisture flux. Therefore, horizontal moisture fluxes at specific altitudes are better for our analysis than the vertical integral of the moisture flux. However, since wind carries moisture away from its source, we expected that wind speed and LMR would also correlate for the higher latitudes (latitude above 45°). It could be that for the higher latitudes, a more significant amount of moisture is present at higher latitudes, explaining why LMR and wind at 10m do not correlate. However, wind speeds at 650 hpa and 750 hpa also do not correlate to LMR for these latitudes (Tab. A2).

Despite the importance of vertical shear in atmospheric moisture tracking models (Van der Ent et al., 2013), we do not find a correlation between local moisture recycling and vertical shear between 650 and 750 hPa. Shear is the friction between air layers that minimizes complete mixing, which for some regions around the world is strongest between 650 and 750 hPa (Dominguez et al., 2016). A possible explanation is that due to its small spatial scale, the temporal scale of LMR is also small, which may prevent the air reaching 700 hPa within the spatial scale of LMR. Furthermore, it is possible that our study design is insufficient to capture the relation between LMR and shear throughout the year over the globe. We aimed for a general analysis to identify the main factors that might influence LMR. A more detailed study that distinguishes between different seasons and isolates different climate zones is necessary to identify more factors that influence LMR as some factors might be more important during a specific season. For example, convection occurs more during summer than during winter, and therefore, might have a stronger correlation with LMR during summer. Besides, some factors are shape and size dependent similar to LMR, while other factors are not dependent on grid cell size and shape. This might cause bias in the results of the Spearman analysis. Furthermore, due to the many interactions within the Earth system and, consequently, between the variables

included in our study, it is impossible to determine the true drivers of LMR. However, the correlations do indicate how changes
in the environment might affect LMR.

**4.2 regional patterns**

To zoom in on the importance of each of the different factors underlying LMR for various areas across the globe, we determined
LMR for the major global biomes (Fig. A8). LMR is highest for the wet tropics (between 0° and 15° north and south) and
montane grasslands and lowest for desert-like biomes in both the Northern and Southern Hemisphere (between 30° and 45°
north and south), confirming the importance of wetness, orography, and latitude. However, in the tropics (between 0° and 15°
latitude), we do not find any correlation between LMR and precipitation, evaporation, wetness, or orography. Possibly, due to
the abundance of water and energy to evaporate, there is LMR under all circumstances, except for when the wind speed is
high. Comparing LMR for each biome between both hemispheres indicates that some of the factors underlying LMR are more
robust than other ones for some biomes. In the Mediterranean biomes, located between 30–40° north and south, air generally
descends due to the Hadley cell circulation. As a result, these biomes are expected to have low LMR. Although we find a low
LMR for the Mediterranean biomes in the Southern Hemisphere, we find a relatively high LMR for the Mediterranean biomes
in the Northern Hemisphere. The Spearman rank analysis indicates that at these latitudes, wind speed correlates with LMR,
which may explain the difference between both hemispheres.

Although LMR is the highest in the wet tropics, we find different results among the various tropical regions (Amazon Basin,
Congo Basin & Southeast Asia). LMR in the Congo Basin exceeds LMR in the Amazon Basin (Fig. 1), despite larger amounts
of precipitation in the Amazon Basin (Hersbach et al., 2020). In the tropics, current deforestation results in drying (Bagley et
al., 2014; Staal et al., 2020), reducing evaporation. As LMR in the Congo Basin exceeds LMR in the Amazon Basin,
deforestation has a relatively large impact on local precipitation in the Congo Basin, suggesting a larger impact on droughts
locally. This is further exacerbated by the fact that the Congo Basin, in comparison with the Amazon Basin, has many small-
scale moisture feedback loops (Wunderling et al., 2022). Unlike LMR, basin recycling is similar for both basins (Tuinenburg
et al., 2020). Suggesting, the impact of deforestation on precipitation in the entire basin is similar for both basins, indicating
both basins would experience similar overall drying. However, drought conditions can also enhance recycling ratios (Bagley
et al., 2014), possibly promoting LMR. Further research is necessary to understand the impact of deforestation on LMR in the
tropics in more detail.

**4.3 The spatial scale of the local moisture recycling ratio**

We study local moisture recycling on a spatial scale of 0.5°, which is approximately 55 km around the equator and 50 km on
average globally for all land cells. Instead of recycling within one grid cell ($r_1$), we studied the recycling of evaporated moisture
within its source grid cell and its 8 surrounding grid cells. Compared to $r_1$, this $r_9$ includes all moisture flows with a length
scale of typically 50 km (0.5°). For $r_1$, moisture flows with a length smaller than 50 km can occur close to the border of grid
cells and therefore, $r_1$ by definition underestimates the actual recycling. These moisture flows are accounted for in $r_9$.

However, defining LMR on a grid scale gives complications. First, the longitudinal distance for a grid cell size decreases with
latitude, resulting in different sizes and shapes, which makes it difficult to compare LMR among all grid cells. For the low-
and mid-latitudes, the variation in grid cell size affects LMR only slightly, as confirmed when LMR for each grid cell was
scaled to a single area (Fig. A9). Therefore, we believe that the variation in grid size causes only a small bias in the statistical
analysis, as the largest fraction of the land surface is at the low- and mid-latitudes, and moisture recycling is less important for
the higher latitudes. However, it should be noted that for similar wind speed, LMR will be lower in smaller grid cells than
larger grid cells. Second, the spatial scale of recycling is strongly dependent on regional differences such as biome type, the
dominating winds, and the proximity to mountains. For instance, with increasing distance to the Andes mountains the median
travelling distance of transpired moisture from the Amazon forest increases (Staal et al., 2018) and for the Ganges basin,
evaporated moisture is blocked by the Himalayas, limiting upward moisture flow and inducing precipitation (Tuinenburg et
al., 2012). Further, precipitation can be triggered by micrometeorological processes (e.g. Knox et al., 2011; Taylor et al., 2012)
making it unknown at what spatial scale moisture recycling is the dominant process for precipitation. Therefore, we believe
that a grid-based approach to systematically study LMR globally is a solid approach to define and study the physical processes
at a spatial scale >50 km through, for instance, the Spearman analysis to study the underlying processes. However, our
definition of LMR is not sufficient to identify processes on a spatial scale smaller than 50 km that might be relevant.
**4.4 Model and definition dependencies**
It is important to note that the typical length scale of moisture recycling, as defined by Van der Ent & Savenije (2011), allows
for a comparison of regional moisture recycling for different regions around the world due to its independence of the region's
size and shape (Fig A10). The typical length scale of evaporated moisture recycling decreases with increasing recycling. It
peaks over deserts and is small over the tropics and mountainous regions (Fig A9), overlapping with the spatial pattern of
LMR. However, this typical length scale does not allow for the quantification of the amount of recycled moisture and therefore,
it is difficult to apply this metric to study the impact of evaporation changes due to land-use change. Therefore, studies that
aim to quantify moisture recycling locally may best use recycling ratios. However, studies that aim to compare recycling
among different regions can best use the typical length scale of recycling.

In this article, we focus on model dependency as we calculated the differences in magnitude of recycling within one grid cell
of 1.5° obtained from output of the UTrack and WAM2-layers models (Link et al., 2020; Tuinenburg et al., 2020). The spatial
patterns are similar, yet the different magnitudes indicate a large model dependency, and, therefore, an uncertainty in moisture
recycling. Furthermore, Van der Ent et al. (2010) calculated recycling within a grid cell of 1.5° for the years 1999–2008 using
WAM2-layers and found a similar spatial pattern with high recycling over mountainous and tropical regions and low recycling

over desert-like regions. These recycling ratios also have a larger magnitude than LMR. However, it is not straightforward to interpret the differences in recycling ratios as both models use different input data (i.e., ERA5 and ERA-Interim). To assess the possible role of the models in causing the difference in moisture recycling, we describe the main differences between the models. First, WAM2-layers calculates the atmospheric moisture recycling on a larger temporal and spatial scale than UTrack, A larger grid cell size and time step increases the likelihood of evaporation and precipitation taking place within the same small amount of time, which might result in an overestimation of recycling within one grid cell. Second, WAM2-layers generates moisture flows using two vertical layers; therefore, strong winds at specific vertical levels will be described in less detail, reducing estimated moisture transport and enhancing estimated moisture recycling within a single grid cell. Differences between $r_{UTrack}$ and $r_{WAM2\text{-}layers}$ are highly visible over mountainous regions where wind experiences relatively strong friction, highly impacting the wind. Finally, different approaches are used to include vertical mixing in the two models. Vertical mixing causes the greatest error in moisture tracking models, but it is unknown to what extent vertical mixing is underestimated (Stohl et al., 2005; Tuinenburg & Staal, 2020).

Besides studies using atmospheric moisture tracking (e.g., Bagley et al., 2014; Keys et al., 2014;Van der Ent et al., 2010), some previous studies used different methods to calculate regional moisture recycling for a specific area, such as isotope measurements (e.g., An et al., 2017) and bulk recycling models (e.g., Burde & Zangvil, 2001). The most common recycling models are modifications of Budyko's model (Budyko, 1974; Burde and Zangvil, 2001), which are 1D or 2D analytical models. These models assume that the atmosphere is completely mixed, meaning that evaporated water directly mixes perfectly with advected water throughout the entire water column. Because of this assumption, first, these models overlook fast recycling, which describes local showers that yield precipitation before the evaporated water is fully mixed. Excluding fast recycling causes models to underestimate terrestrial moisture recycling for some regions (e.g., Amazon Basin) (Burde et al., 2006b). Second, these models ignore the influence of vertical shear, which causes a significant error (Dominguez et al., 2020). Our method minimises the errors due to fast recycling and vertical shear because of two model aspects. First, at each time step, each parcel has a small chance of getting mixed, causing each parcel to move approximately once in the vertical direction every 24 hours, additional to the displacement based on reanalysis data of vertical winds. This process minimizes complete mixing and reduces the error due to shear and fast recycling. Second, the error due to fast recycling also becomes smaller because lower atmospheric levels contribute more to the total precipitation than higher levels due to the skewed vertical moisture profile. WAM2-layers accounts for vertical shear as it models two vertical atmospheric layers of which the interface is located at the height at which shear typically occurs. These two layers are both completely mixed and therefore, compared to bulk models, WAM2-layers better represents the distribution of moisture throughout the atmospheric column. As an alternative method, moisture flows can be calculated on a smaller time step to increase the interactions between different wind components, resulting in a better representation of turbulence (Keune et al., 2022). Despite the error reduction, the representation of fast recycling in UTrack should be studied in more detail, as fast recycling is expected to influence LMR significantly.

LMR is calculated as a ten-year average. This period of ten years might miss multi-year climate variability such as the El Niño
Southern Oscillation and the North Atlantic Oscillation. The time series of atmospheric moisture connections provided by Link
et al. (2020) allowed to study inter-annual variation in relatively local recycling. This shows that recycling is dependent on
multi-year atmospheric phenomena. During the major El Niño event of 2015-2016, the northeast of South Africa had a lower-
than-average local recycling ratio (Fig. A11) for 2015. This pattern coincides with the impact of wetness during El Niño years,
consistent with the hypothesis that wetness enhances LMR. Furthermore, strong events such as heat waves and droughts might
affect the multi-year annual mean. For example, we clearly find lower recycling over Russia during 2010, which may relate to
the 2010 heatwave in eastern Europe and Russia. Overall, for these multi-year and strong events we find that, for regions that
face wetter-than-normal conditions, LMR is enhanced, and for regions that face drier-than-normal conditions, LMR is reduced.
Hence, drought events might result in a decrease in LMR as seen for the 2010 heat wave event in Europe and Russia. However,
not for all inter-annual climate variability modes we find a clear impact on moisture recycling. It may be that these phenomena
do not affect wetness throughout the entire year, and therefore, annual means might not represent them well.

**4.5 Implications/applications of LMR**

LMR could be applied in the field of water management. The spatial pattern of LMR shows some overlap with global
agricultural water management (Molden, 2007; Salmon et al., 2015). Generally, the tropics have a high LMR and agriculture
is mainly rainfed (Salmon et al., 2015; Costa et al., 2019), indicating that these agricultural regions are self-dependent to some
extent regarding precipitation . Also, agriculture in the Mediterranean Basin and South Australia is mainly rainfed. For semi-
arid regions that dependent on rainfed agriculture, changes in precipitation may have a significant impact (Keys et al., 2016).
LMR in the Mediterranean basin exceeds LMR in southern Australia, indicating that a larger fraction of evaporated moisture
returns locally. Thus, when evaporation is maintained in the Mediterranean Basin, part of the precipitation will sustain here,
which holds to a lesser extent for southern Australia. Besides LMR (i.e., local evaporation recycling), local precipitation
recycling can help to understand the precipitation dependence on local evaporation for each region. Irrigated agriculture is
important in India and China (Salmon et al., 2015; Döll and Siebert, 2002), which are regions with a relatively low LMR,
indicating that only a small amount of the evaporated moisture returns as precipitation locally. For irrigated agriculture in
regions that are characterized by a high LMR, a relatively large amount of the evaporated water returns to its source, which
reduces the amount of water that is necessary for irrigation. Terrestrial evaporation is an important source for precipitation and
freshwater availability (Keune and Miralles, 2019). Therefore, spatial planning using LMR might improve agricultural water
management.
Global climate change likely affects atmospheric moisture connections due to changes in atmospheric dynamics. For example,
due to global warming, tropical atmospheric circulation may weaken (Vecchi et al., 2006), and the Hadley cells may move
poleward (Shaw, 2019), which will affect the updraft and downdraft of air around the globe, which we found to be important
processes underlying LMR. Furthermore, climate change has different opposing impacts on storm tracks which have an

important role in moisture transport by transporting latent heat poleward (Shaw et al., 2016). Furthermore, in a warmer climate continental recycling is predicted to decrease and precipitation over land would be more dependent on evaporation over the ocean (Findell et al., 2019). However, our study does not account for any impacts of climate change. As our results indicate that wetness and convection enhance LMR, LMR may change due to, for example, drying and wetting of regions, changes in Hadley cell circulation, and circulation in the tropics. Furthermore, climate change enhances the risk of droughts (Rasmijn et al., 2018; Teuling, 2018) and LMR might be used to study drought resilience globally. As for a high LMR a local drought might drastically impact the local water cycle.

We expect that LMR can be helpful also in other ways. Specifically, we expect the concept of LMR can be used to study how changes in evaporation, due to for example afforestation, affect the local water cycle beyond merely a loss of moisture. However, besides evaporation, land-use changes also influence the energy balance and other factors that might alter the atmospheric moisture connections and thus, LMR. Using future land use scenarios as input for moisture tracking models, it will be possible to study the impact of land-use changes on atmospheric moisture connections. However, future scenarios often include other changes besides land use, which makes it possible to study the changes of land use specifically. However, LMR can help us better predict the impact of land cover changes on the local water cycle. It might help us identify regions where reforestation would not cause local drying due to enhanced evaporation (Hoek van Dijke et al., 2022; Tuinenburg et al., 2022). Overall, LMR gives us better insight into the atmospheric part of the local water cycle and terrestrial evaporation as a source for local freshwater availability.

**5 Conclusions**

We calculated the local moisture recycling ratio (LMR) from atmospheric moisture connections at a spatial scale of 0.5°. LMR is the fraction of evaporated moisture that precipitates within a distance of 0.5° (typically 50 km) from its source. On average, 1.7% (st.dev. = 1.1%) of global terrestrial evaporation returns as precipitation locally, with peaks of approximately 6%. LMR peaks in summer and in wet and elevated regions. We find a correlation between LMR and orography, precipitation, wetness, convective available potential energy, and wind suggesting these variables might affect LMR. In addition, latitude correlates with LMR, which likely indicates the importance of the ascending air and descending air related to the Hadley cell circulation. Furthermore, by comparing LMR calculated using different models we found that the spatial pattern of LMR is not model-dependent, yet, the magnitude of LMR is strongly dependent on the model. LMR defines the local impacts of enhanced evaporation on precipitation and thus its role as a source for local freshwater availability. Therefore, LMR can be used to evaluate which locations may be suitable for regreening without largely disrupting the local water cycle.

**Appendix A**

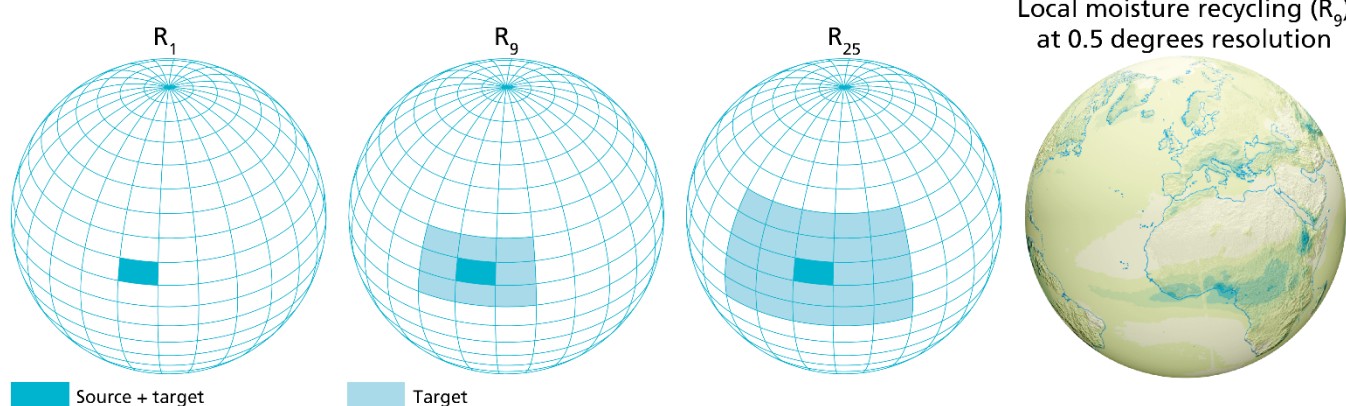

Source + target   Target

Figure A1. Three definitions of the local moisture recycling ratio (LMR) from left to right: $r_1$ describes the fraction of evaporated
moisture that returns as precipitation in its source grid cell, $r_9$ describes the fraction of evaporated moisture that returns as
precipitation in its source grid cell and 8 neighbouring grid cells, and $r_{25}$ describes the fraction of evaporated moisture that returns
as precipitation in its source grid cell and 24 neighbouring grid cells. LMR is calculated on a spatial scale of 0.5° and the first three
plots do not have a similar resolution. The plot on the right shows LMR on a spatial scale of 0.5° which is the resolution at which we
calculate all definitions ($r_1$, $r_9$ and $r_{25}$).

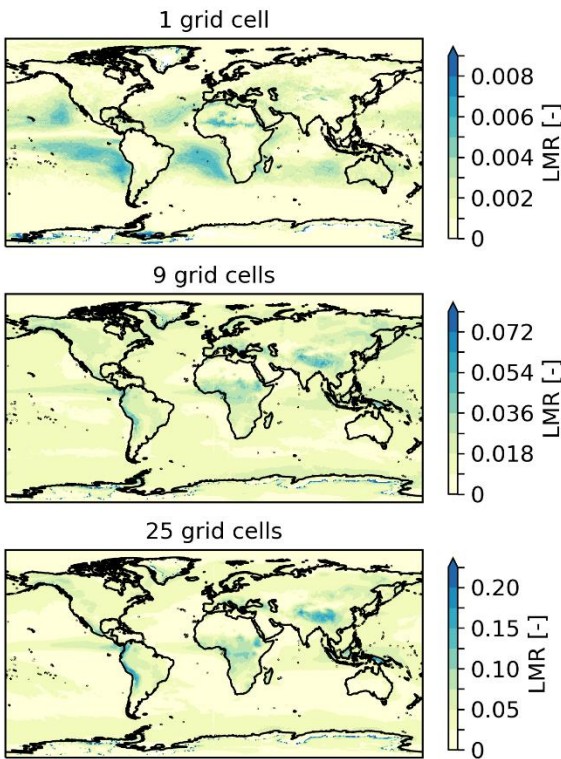


Figure A2. 10-year climatology (2008–2017) of the three definitions of the local moisture recycling ratio (LMR). The top panel indicates the fraction of evaporated moisture that precipitates within its source grid cell ($r_1$), the middle panel shows the fraction of evaporated moisture that precipitates within its source grid cell and its 8 neighbouring grid cells ($r_9$), and the lower panel shows the fraction of evaporated moisture that precipitates within its source grid cell and its 24 neighbouring grid cells ($r_{25}$).

Table A1: Defined classes for spearman rank correlation analysis.

| *Class* | Latitude ranges |
|---|---|
| *1* | -15°:15° |
| *2* | -30°:-15° and 15°:30° |
| *3* | -45°:-30° and 30°:45° |
| *4* | -60°:-45° and 45°:60° |
| *5* | 60°:75° |
| *6* | 75°:90° |

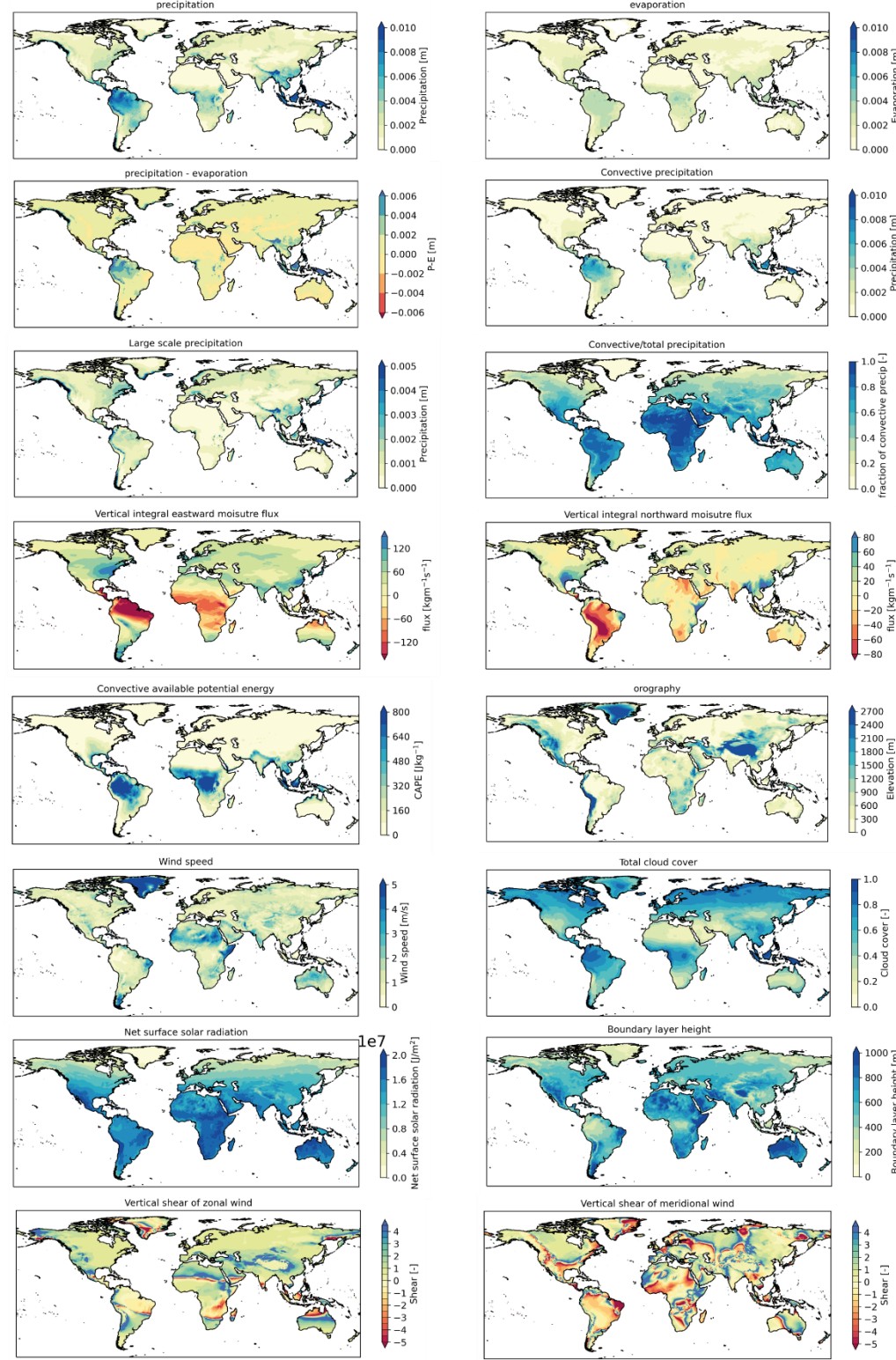

**Figure A3. Global 10-year climatology (2008–2017) of (from top to bottom and left to right) precipitation, evaporation, precipitation**
**– evaporation, convective precipitation, large-scale precipitation, fraction of convective precipitation, vertical integral of moisture**
**flux in eastward direction, vertical integral of moisture flux in northward direction, CAPE, orography, vertical shear (between 650**
**and 750 hPa) of zonal wind, and vertical shear (between 650 and 750 hPa) of meridional wind.**

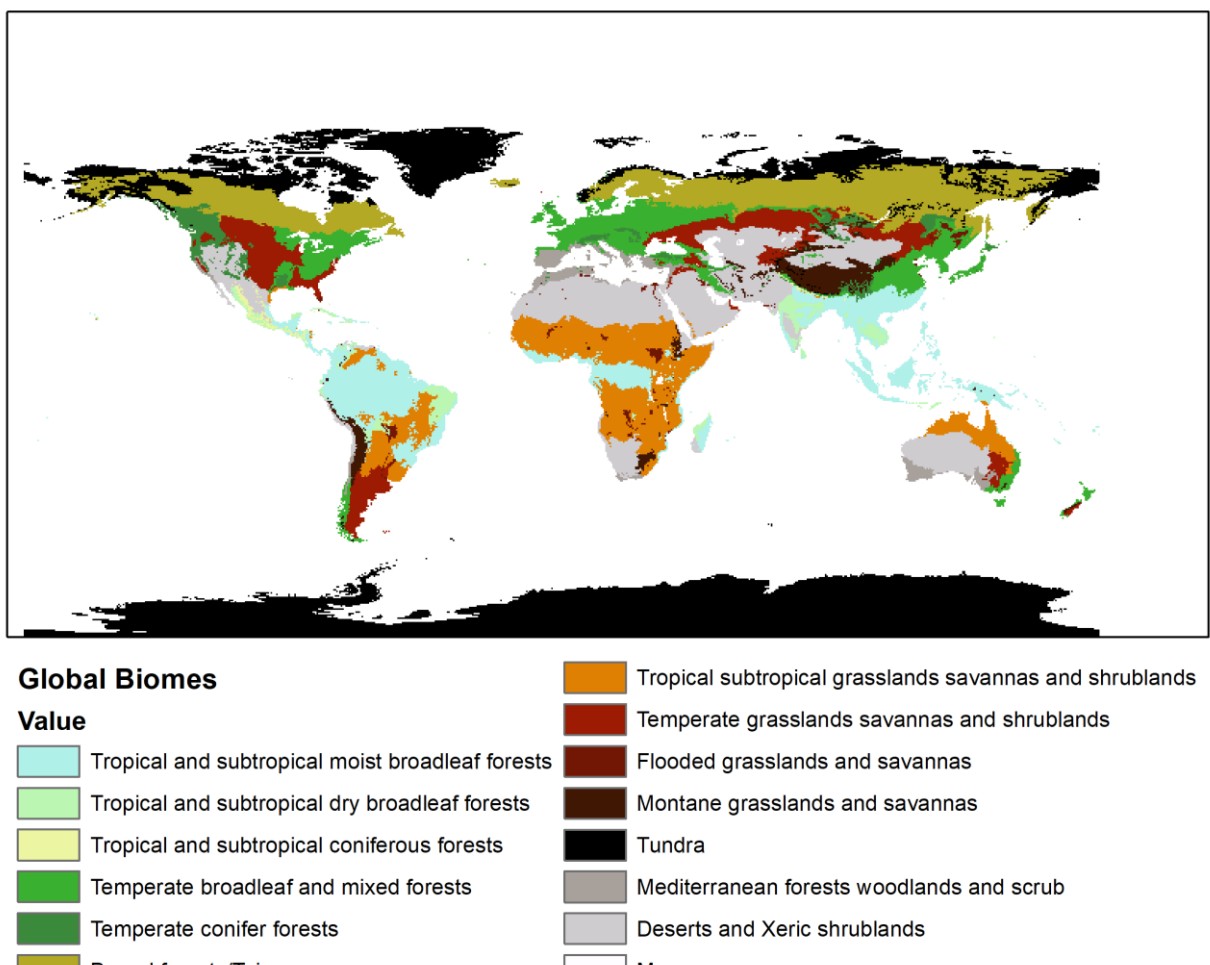

**Figure A4. Major global biomes Ecoregions 2017 (https://ecoregions.appspot.com/) ©NASA Terra Metrics, @Google INEGI**
**Imagery**

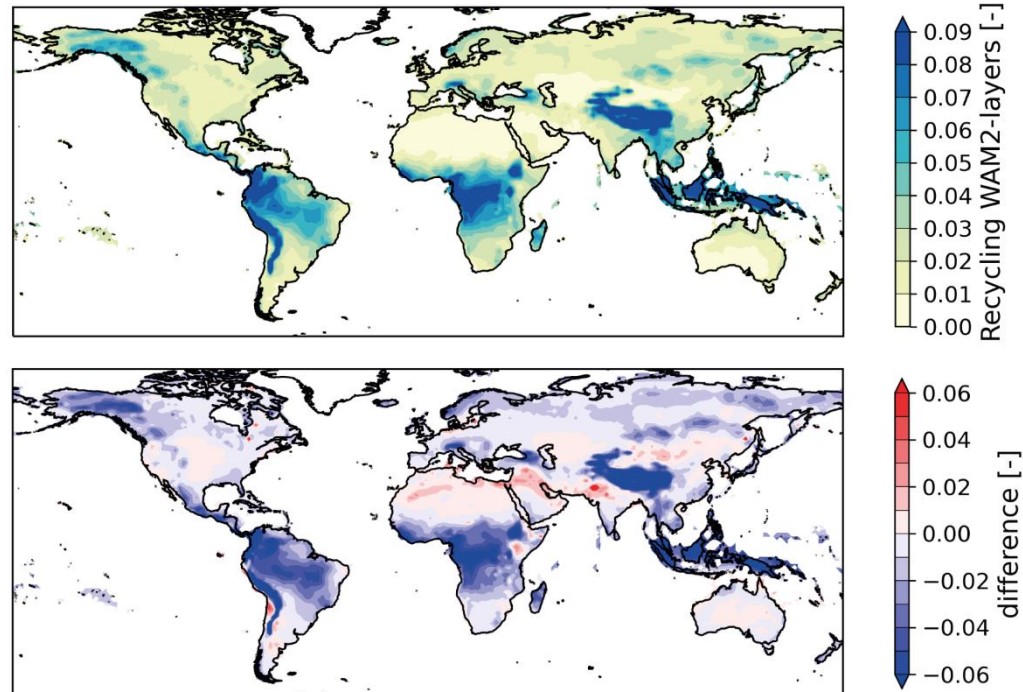

**Figure A5. The 10-year climatology (2008-2017) of the recycling within one grid cell of 1.5° calculated with the dataset by Link et**
**al. (2020), i.e., the output from the Eulerian moisture tracking model WAM2-layers (top) and the difference with the The 10-year**
**climatology (2008-2017) of the recycling within one grid cell of 1.5° calculated with the dataset by Tuinenburg et al. (2020).**

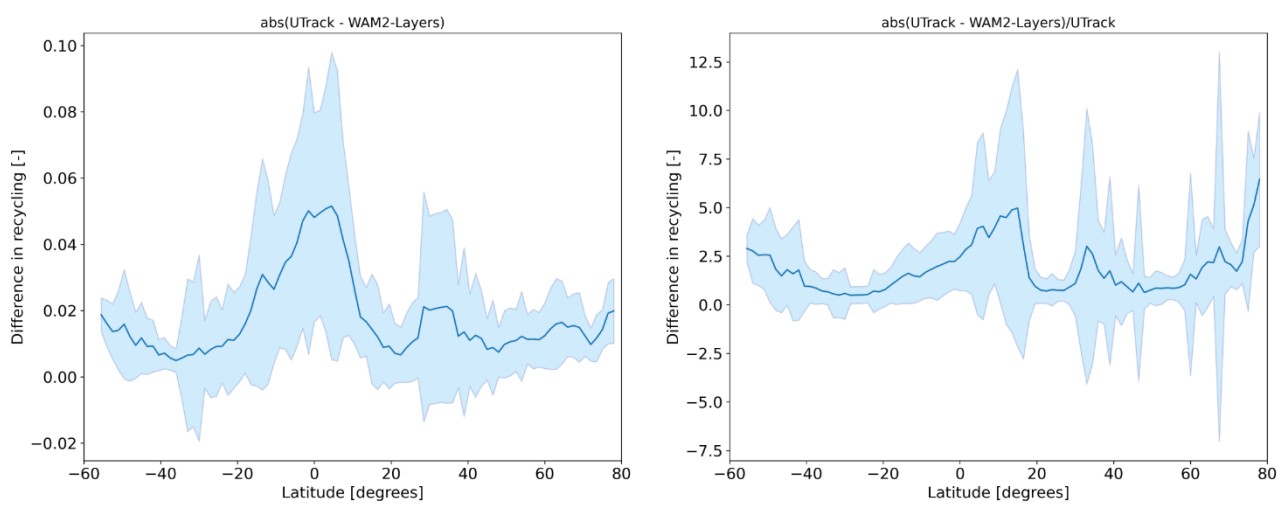


**Figure A6. The zonal mean of the absolute difference (left) and relative difference (right) between $r_{UTrack}$ and $r_{WAM2\text{-}layers}$ (calculated**
**as $r_{UTrack}$ minus $r_{WAM2\text{-}layers}$, indicated by the blue line) and its standard deviation (blue area).**

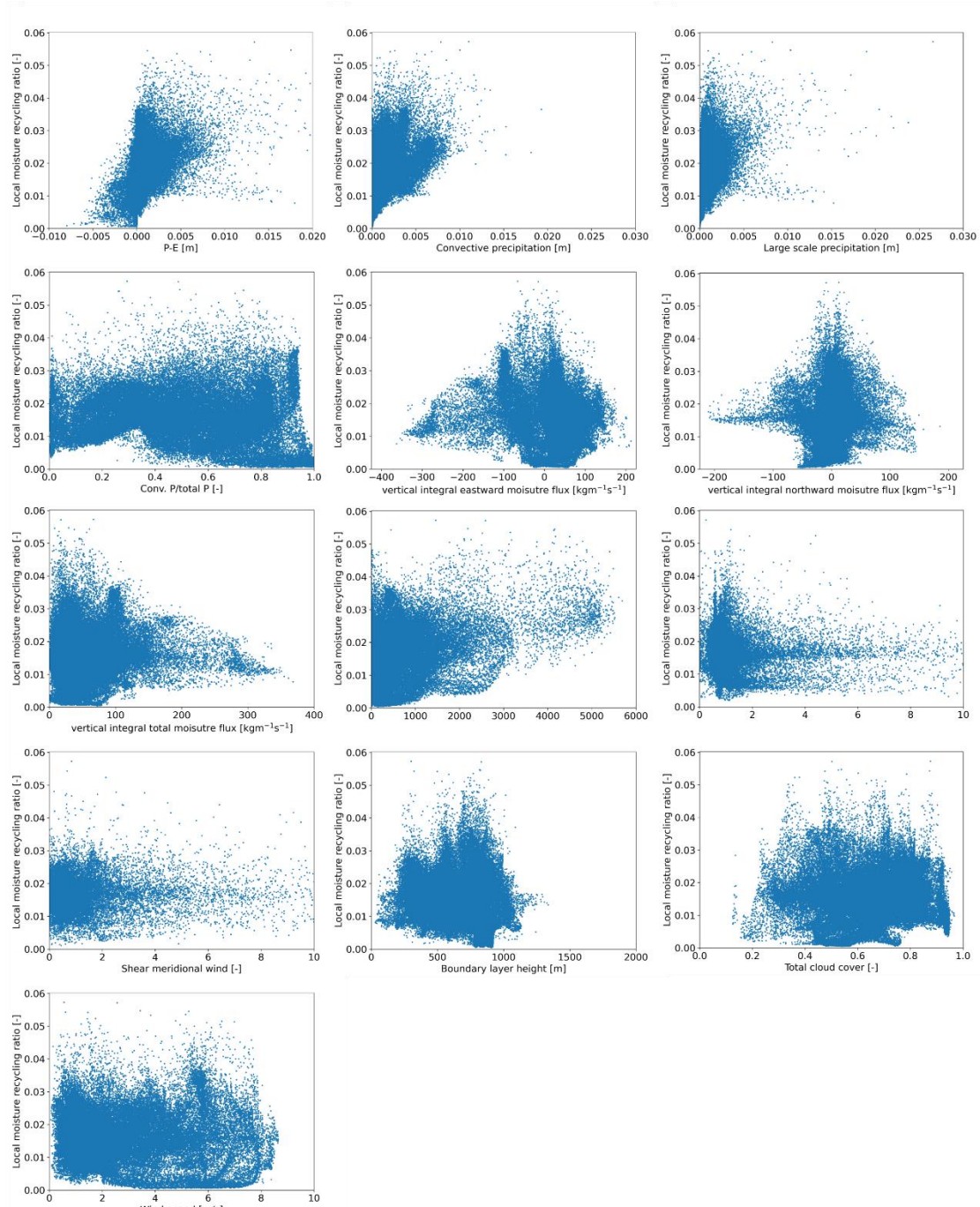


**Figure A7. Scatter plots of the 10-year climatology (2008–2017) of the local moisture recycling ratio and (from top to bottom and left to right) precipitation – evaporation, convective precipitation, large-scale precipitation, fraction of convective precipitation, vertical integral of moisture flux in eastward direction, vertical integral of moisture flux in northward direction, orography, vertical shear (between 650 and 750 hPa) of zonal wind, vertical shear (between 650 and 750 hPa) of meridional wind, boundary layer height, total cloud cover, and wind speed. Each scatter represents one grid cell.**

**Table A2. Spearman rank correlation coefficients for additional variables at different latitude classes. '*' indicates a significant correlation (p<0.05) and moderate and strong relations (ρ>0.4) are emboldened. The classes including latitudes between 0° and 60° include grid cells of the Northern Hemisphere and Southern Hemisphere. The classes including latitudes exceeding 60° include grid cells of the Northern Hemisphere only.**

| Variables | Spearman rank correlation coefficient | | | | | |
|---|---|---|---|---|---|---|
| | 0°-15° | 15°-30° | 30°-45° | 45°-60° | 60°-75° | 75°-90° |
| Total cloud cover and wind speed | **-0.58** | **-0.41** | -0.23 | 0.08 | 0.16 | **-0.51** |
| Large-scale precipitation and wind speed | -0.30 | **-0.46** | -0.37 | 0.06 | 0.11 | -0.28 |
| Convective precipitation and wind speed | **-0.63** | **-0.50** | -0.33 | -0.13 | **-0.41** | **-0.61** |
| Total cloud cover and precipitation | **0.85** | **0.92** | **0.76** | **0.58** | -0.08 | **0.46** |
| Total cloud cover and convective precipitation | **0.85** | **0.90** | **0.63** | 0.23 | -0.09 | **0.67** |
| Total cloud cover and large-scale precipitation | **0.71** | **0.90** | **0.81** | **0.70** | -0.02 | **0.43** |
| LMR and wind speed at 650 hpa | 0.26 | -0.18 | -0.37 | -0.16 | -0.15 | -0.27 |
| LMR and wind speed at 750 hpa | -0.09 | 0.023 | -0.39 | -0.19 | -0.09 | -0.31 |

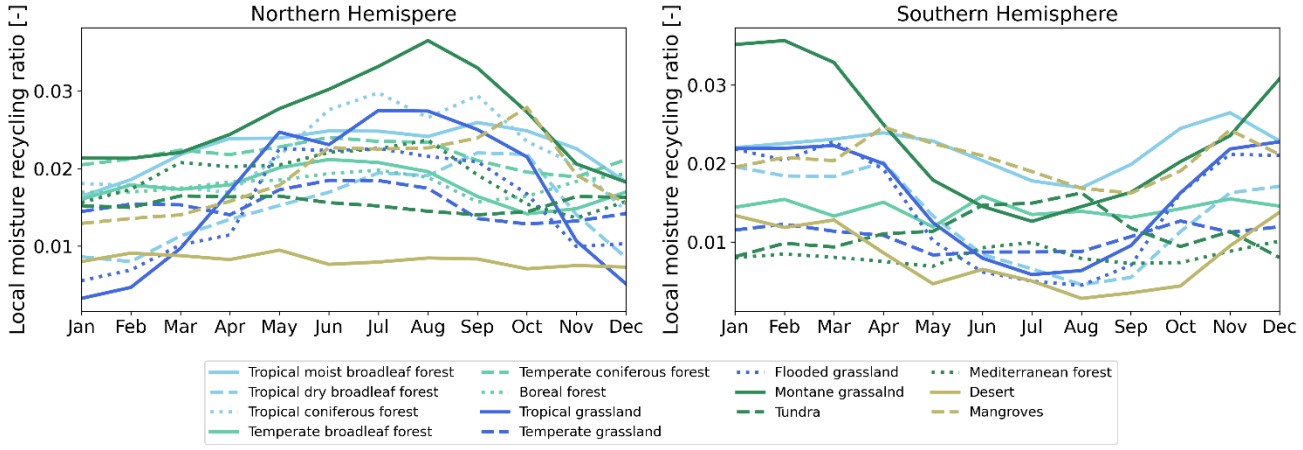

**Figure A8. Time series of the local moisture recycling ratio for global biomes on the Northern (left) and Southern (right) Hemispheres. The plots show the 10-year climatology (2008–2017).**

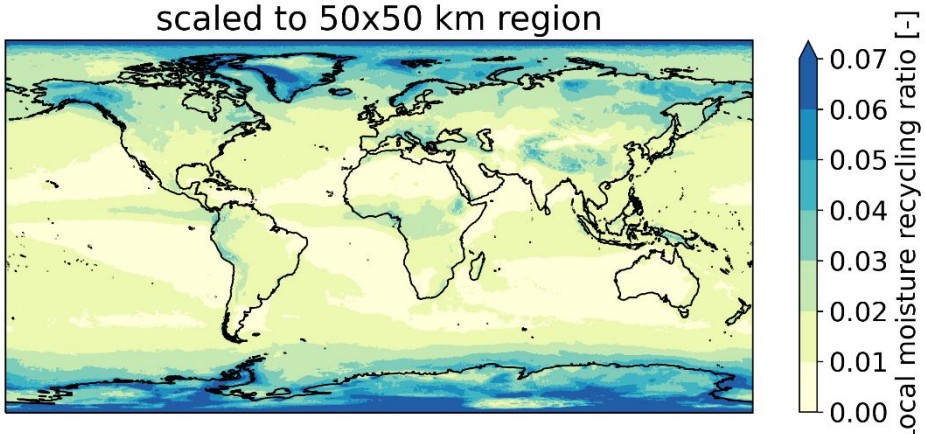

500

**Figure A9: The local moisture recycling ratio scaled to a grid cell size of 50 km x 50 km. The plot shows the 10-year climatology (2008-2017). We divided the original local moisture recycling ratio by the area of the grid cell and multiplied it with 2500 km²**

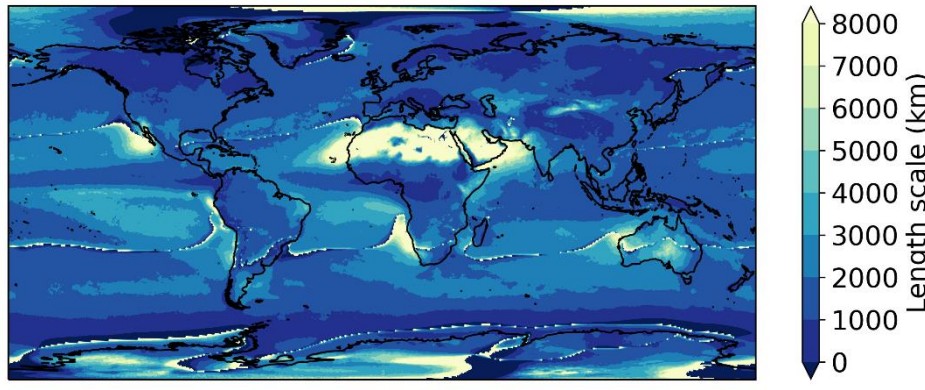

503

**Figure A10: Evaporation recycling length scale as defined by Van der Ent and Savenije (2011) for each grid cell of 0.5°x0.5°. The plot shows the average of 2008-2017.**

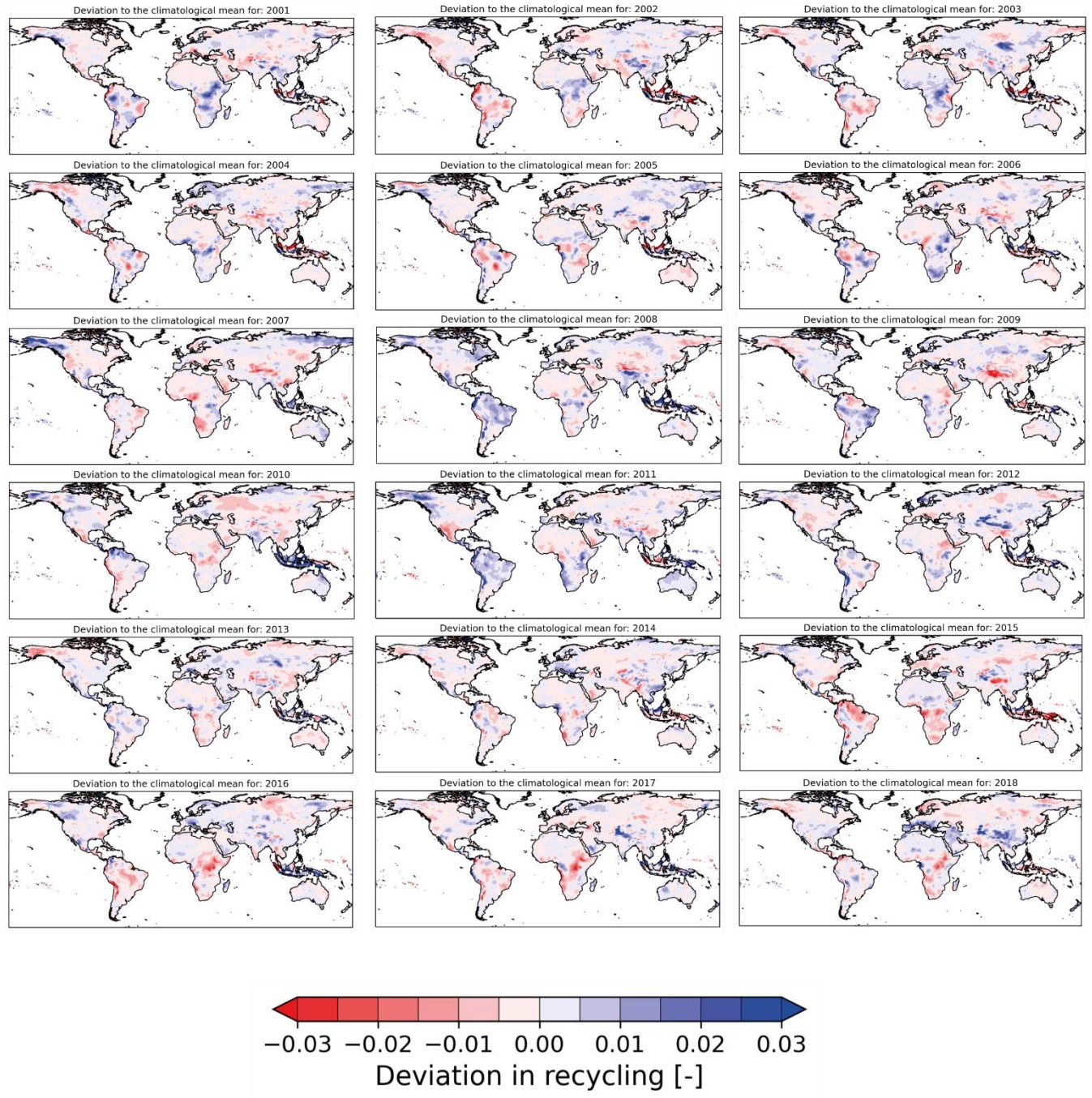

Figure A11. Inter-annual variation of recycling within a single grid cell of 1.5° between 2001-2018. Each plot shows the difference between annual averaged recycling and the climatological mean of recycling. Data obtained from Link et al. (2020).

## Code availability

The code that was used to calculate the local moisture recycling ratio and to plot the local moisture recycling ratio is provided on GitHub (https://github.com/jtheeu/LocalMoistureRecycling, last access 28 February 2023).

## Data availability

The local moisture recycling ratios are available from the Zenodo archive at 0.5 and 1.5 degrees resolution (https://doi.org/10.5281/zenodo.7684640).

The atmospheric moisture connections from Tuinenburg et al., (2020) are available from the PANGAEA archive at 0.5 and 1.0 degrees resolution (https://doi.pangaea.de/10.1594/PANGAEA.912710).

The atmospheric moisture connections from Link et al., (2020) are available from the PANGAEA archive at 1.5 degrees resolution (https://doi.pangaea.de/10.1594/PANGAEA.908705).

## Author contributions

JT designed the study with contributions from all authors. JT carried out the research. JT wrote the first draft of the manuscript in close collaboration with AS. All authors contributed to the discussion and the final version of the manuscript.

## Acknowledgements

We want to thank Patrick Keys, Ruud van der Ent and the anonymous reviewer for commenting on earlier versions of this manuscript. This work was performed in the cooperation framework of Wetsus, European Centre of Excellence for Sustainable Water Technology (www.wetsus.eu). Wetsus is co-funded by the Dutch Ministry of Economic Affairs and Climate Policy, the Northern Netherlands Provinces the Province of Fryslân. The authors would like to thank the participants of the natural water production theme for the fruitful discussions and financial support. AS acknowledges support from the Talent Programme grant VI.Veni.202.170 by the Dutch Research Council (NWO). OT acknowledges support from the research programme Innovational Research Incentives Scheme Veni (016.veni.171.019), funded by the Dutch Research Council.

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
