# Peer review of "Local moisture recycling across the globe"

_EGUsphere, 2022_

## Community Comment (CC3)

Dear Dr. van der Ent,

Thank you for your time and the comments you provided on our manuscript: 'Local moisture recycling across the globe'. Your comments are of great value to improve our manuscript. Below we respond to each point separately to discuss how we will implement them.

**what's the reason to call your metric 'local' recycling? Isn't it just regional recycling for grid cells of 0.5 arcdegree x 0.5 arcdegree?**

Currently, in literature a similar concept (regional recycling) is used to describe recycling over areas with varying size. Such an area is often related to a peninsula, catchment, etc. Our metric is different from this as we provide information on a more specific and smaller scale namely 0.5 degrees. This is currently the best resolution available for such data with global coverage. We refer to our new metric as the local moisture recycling ratio, first, to highlight the difference with previous studies on regional recycling. Second, this resolution approaches a length scale where local processes have relevant contributions. Feedbacks between the land surface and atmosphere at this spatial scale are relevant for regreening projects. So, we decided to call our metric local recycling as we see potential to use it to study local feedbacks.

**I find the novelty somewhat overstated. As far as I understand the novelty is simply the fact that you calculate regional recycling on a higher resolution grid than other global studies, but the conceptual calculation is very old (see reference list in van der Ent and Savenije (2011) for example).**

We see there are some similarities to the work done by Van der Ent and Savenije (2011), yet we believe our work is novel. In our paper we assess the impact of the spatial scale within which the moisture recycles by calculating three different types of local moisture recycling. Furthermore, we assess potential drivers of local moisture recycling and by doing so we contribute to the physical understanding of local recycling. However, we will include the work from Van der Ent and Savenije (2011) in our introduction and we will discuss in more detail how our work compares to the work done by Van der Ent and Savenije. Furthermore, we will explain in more detail how our analyses add to the current knowledge, i.e., we can better understand the spatial patterns in local recycling by assessing its drivers. Additionally, the previous work was conducted using output from a different moisture tracking model (i.e., WAM2-layers), which assumes complete mixing of evaporated moisture within two atmospheric layers. For short time scales complete vertical mixing might not be realistic. The UTrack model distributes the evaporated moisture along the vertical moisture profile and therefore, might be more suitable for analyses on a smaller scale. However, we are thankful for this comment which helps us specifying the novelty of our research better.

The following points will be addressed after the last point.

**A gridcell of 0.5 arcdegree in let's say Stockholm is two times smaller in area than a gridcell around the equator**

**moreover, that same grid cell is 3 times smaller in length in east-west direction, and, therefore the dominant wind direction is rather influential on its value.**

**In other words, the regional recycling metric or LMR is scale and shape dependent and as such its values cannot be compared from region to region.**

**The search for a relation between LMR and other quantities that do not suffer from the scale and shape dependency (precipitation, evaporation, CAPE, biomes etc. ) is therefore fundamentally skewed.**

**In Van der Ent and Savenije (2011) I had a suggested alternative metrics, which actually have local meaning for the recycling process, which are the local length scale of precipitation recycling and the local length scale of evaporation recycling. Surely these also rely on a few assumptions, but they do not suffer (or at least to a much more limited extent) from the scale and shape dependency. Please consider this approach or think of a better way to make your metrics scale and shape independent.**

We see that the spatial scale affects moisture recycling ratios, and we believe this is a very useful comment that will help us to improve the quality of our manuscript. To assess this effect for our results, we scaled the local moisture recycling ratio of each grid cell to an area of 50 km x 50 km (see Figure R1). The relatively large difference between local recycling and scaled local recycling at high latitudes indicates that the local moisture recycling is more uncertain for higher latitudes. For the rest of the globe, we find that the general patterns of local recycling we describe in our manuscript are consistent with and without scaling. The terrestrial surface at lower and mid latitudes are most important for moisture recycling and as the pattern of the scaled and non-scaled local recycling ratio are similar here, we believe the grid cell size causes only a small bias here.

Besides scaling, Van der Ent and Savenije (2011) presented another metric to describe the local moisture recycling, namely, the length scale of evaporation recycling (we will refer to this metric as length scale). They found that this metric scales with their definition of local recycling ratio and has a value typically in the order of 1000 km globally. We did calculate this length scale for our data (see Figure R2), and its patterns are similar to the patterns we found for local recycling with large values (i.e., small length scales) over tropics and mountainous regions and small values (i.e., large length scales) over desert areas. Similar to the result from Van der Ent and Savenije (2011) we find length scales in the order of 1000 km. Although it is an important metric, we believe that it is more difficult to apply it to, for instance, the impact of land use change on precipitation locally. For this, one needs to determine the amount of rain that recycles locally, and length scale does not quantify this whereas the local moisture recycling ratio does. Yet, to apply local moisture recycling locally, we believe it is important to better understand the local moisture recycling ratio first, and with our study we add to its understanding. In the discussion of our paper, we will discuss the differences between the local moisture recycling ratio and this length scale to allow readers to better assess what metric to use when addressing research questions related to moisture recycling as we do see value in using the length scale for research questions related to non-local effects.

Concerning the comment on the impact of wind direction, we fully agree that the dominant wind direction affects the value of LMR. We can imagine that especially on higher latitudes, where variation between grid cell shape and size is relatively large and the difference between the zonal and meridional length of the grid cell is large, wind direction might have a strong impact on local recycling as you indeed pointed out. We will clarify the impact of dominant wind direction in our discussion to create awareness of this effect amongst the readers.

Finally, you state that the analysis in which we study correlations between the local recycling and different variables is fundamentally skewed due to the scale and shape dependency, which some of the other variables don't have. We agree that the set-up of our study may result in skewed output, which is intrinsic to our data. Therefore, there is no perfect method to make a comparison between recycling ratios in different regions to study the correlation between the local moisture recycling

ratio and other variables. However, to address research questions related to quantifying local hydrology, the local moisture recycling ratio is useful and currently no other metric is available to quantify this globally. A comparison of the recycling ratio among different grid cells is difficult to physically interpret (Van der Ent and Savenije, 2011). Therefore, we need to gain a better understanding of the local moisture recycling ratio. We build this understanding by identifying some of its drivers. However, based on this comment we plan to account for this effect by conducting another analysis in which we classify the data based on latitude and calculate correlation coefficients for the data in these different classes. The grid cell sizes within each class will then be more comparable to minimize the skewness of the analysis. Furthermore, because mainly the higher latitudes are skewed, we excluded Antarctica from our analysis.

To summarize, we will use your comments to address the issue of scaling in more detail in our manuscript. This allows readers to put our results better in perspective. To support this, we will add figures R1 and R2 in the appendix of our manuscript. We would like to thank you for your constructive feedback as it is valuable for improving our manuscript.

On behalf of all authors,

Jolanda Theeuwen

[Figure]

Figure R1: The local moisture recycling ratio scaled to a grid cell size of 50 km x 50 km. The plot shows the average of 2008-2017. We divided the original local moisture recycling ratio by the area of the grid cell and multiplied it with 2500 $km^2$.

[Figure]

[Figure]

Figure R2: Evaporation recycling length scale as defined by Van der Ent and Savenije (2011) for each grid cell of 0.5x0.5 degrees. The plot shows the average of 2008-2017.

---

## Author Response (AR1)

Dear dr. Alexander Gruber,

Thank you for giving us the opportunity to revise our manuscript "Local moisture recycling across the globe" for *Hydrology and Earth System Sciences*. We would also like to thank dr. Ruud van der Ent, dr. Patrick Keys and the anonymous reviewer for their valuable feedback on our manuscript. Their comments helped us to improve our manuscript. Based on these comments we included additional steps in our methodology. For example, first, we used the dataset by Link et al. (2020) to determine model dependency of recycling within one grid cell of 1.5 degrees. Second, we included more variables in our Spearman rank correlation analysis. Finally, to account for differences in grid cell size in the Spearman rank correlation, we classified our data based on latitude. Below, we discuss all comments of both the reviews and the community comments and describe how we implemented them in our revised manuscript. We do this for each review and comment separately. Our response to each comment is presented in blue. When we refer to the line numbers where we made changes, we refer to the line numbers in the document including the tracked changes.

Besides the revisions suggested by the reviewers we made a few other changes. First, we made some small changes to the methods which slightly affects LMR, i.e., to calculate the 10-year climatology of LMR, for each month we weighted the multi-year (2008-2017) monthly LMR by multi-year monthly evaporation in the same period (see lines 162-165). Second, we moved the first paragraph of the results to the methods, and we moved Fig. 1 to the appendix, as we now consider it supplementary to the methods. Third, the conceptual figure in the discussion has been replaced with a clearer and more attractive figure. Fourth, in our discussion we rewrote the paragraph in which we discuss LMR in different biomes. Using the output from the Spearman rank correlation analysis for which we classified our data based on latitude, we discuss the relevance of drivers at different biomes and how that relates to the correlation between LMR and different variables at similar latitude.

We hope that our manuscript is now considered suitable for publication in *Hydrology and earth System Sciences*. In case of further questions or requests, please let me know.

On behalf of all authors,

Yours sincerely,

Jolanda Theeuwen

**Comment dr. Ruud van der Ent**

**what's the reason to call your metric 'local' recycling? Isn't it just regional recycling for grid cells of 0.5 arcdegree x 0.5 arcdegree?**

Currently, in literature a similar concept (regional recycling) is used to describe recycling over areas with varying size. Such an area is often related to a peninsula, catchment, etc. Our metric is different from this as we provide information on a more specific and smaller spatial scale namely 0.5 degrees. This is currently the best resolution available for such data with global coverage. We refer to our new metric as the local moisture recycling ratio, first, to highlight the difference with previous studies on regional recycling. Second, this resolution approaches a spatial scale where local processes have relevant contributions. For example, feedbacks between the land surface and atmosphere at this spatial scale are relevant for regreening projects. So, we decided to call our metric local recycling as we see potential to use it to study local feedbacks. We describe the

relevance of studying moisture recycling at a smaller scale in the fourth paragraph of the introduction (lines 61-66) of our revised manuscript.

**I find the novelty somewhat overstated. As far as I understand the novelty is simply the fact that you calculate regional recycling on a higher resolution grid than other global studies, but the conceptual calculation is very old (see reference list in van der Ent and Savenije (2011) for example).**

We agree that there are some similarities to the work done by Van der Ent and Savenije (2011), yet we believe our work is mostly novel. In our paper we assess the impact of the spatial scale on moisture recycling by calculating three different types of local moisture recycling. Furthermore, we assess potential drivers of local moisture recycling and by doing so we contribute to the physical understanding of local recycling. However, we now incorporated the work from Van der Ent and Savenije (2011) in our introduction (lines 26 and 51-57) and we discuss in more detail how our work compares to the work done by Van der Ent and Savenije (lines 378-390). Furthermore, we explain in more detail how our analyses add to the current knowledge, i.e., we can better understand the spatial patterns in local recycling by assessing its drivers. Additionally, the previous work was conducted using output from a different moisture tracking model (i.e., WAM2-layers), which assumes complete mixing of evaporated moisture within two atmospheric layers. For short time complete vertical mixing might not be realistic. The UTrack model distributes the evaporated moisture along the vertical moisture profile and therefore, might be more suitable for analyses on a smaller scale. In our revised manuscript, we added a comparison of moisture recycling within one 1.5-degree grid cell obtained from the output of UTrack and the output of WAM2-layers. In our introduction we shortly describe WAM2-Layers and refer to relevant literature for more information (lines 51-54). Finally, we discuss the differences between the different models and how they might affect the results in lines 392-408. However, we are thankful for this comment which helps us specifying the novelty of our research better.

The following points will be addressed after the last point.

**A grid cell of 0.5 arcdegree in let's say Stockholm is two times smaller in area than a grid cell around the equator**

**moreover, that same grid cell is 3 times smaller in length in east-west direction, and, therefore the dominant wind direction is rather influential on its value.**

**In other words, the regional recycling metric or LMR is scale and shape dependent and as such its values cannot be compared from region to region.**

**The search for a relation between LMR and other quantities that do not suffer from the scale and shape dependency (precipitation, evaporation, CAPE, biomes etc. ) is therefore fundamentally skewed.**

**In Van der Ent and Savenije (2011) I had a suggested alternative metrics, which actually have local meaning for the recycling process, which are the local length scale of precipitation recycling and the local length scale of evaporation recycling. Surely these also rely on a few assumptions, but they do not suffer (or at least to a much more limited extent) from the scale and shape dependency. Please consider this approach or think of a better way to make your metrics scale and shape independent.**

Indeed, the spatial scale affects moisture recycling ratios, and we believe this is a very useful comment that will help us to improve the quality of our manuscript. We included this issue in the

introduction of our revised manuscript (lines 53-54) to make the readers aware of it. To assess this effect for our results, we present additional results in which we scaled the local moisture recycling ratio of each grid cell to an area of 50 km x 50 km (see Figure R1). For the high latitudes, scaling LMR results in a large difference. For the low and mid-latitudes, we find that the general patterns of local recycling we describe in our manuscript are consistent with and without scaling. The terrestrial surface at lower and mid latitudes are most important for moisture recycling and as the pattern of the scaled and non-scaled local recycling ratio are similar here, we believe the grid cell size causes only a small bias here. We shortly discuss this in our revised manuscript (lines 361-366) and added a figure showing scaled LMR to our additional Figures (Figure A9).

Besides scaling, Van der Ent and Savenije (2011) presented another metric to describe the local moisture recycling, namely, the length scale of evaporation recycling (we will refer to this metric as length scale). They found that this metric scales with their definition of local recycling ratio and has a value typically in the order of 1000 km globally. To better embed our work in literature and indicate its relevance, we mention this typical length scale in the introduction of our revision (lines 64-59). We now also calculate this length scale for our data (see Figure A10), and its patterns are similar to the patterns we found for local recycling with large values (i.e., small length scales) over tropics and mountainous regions and small values (i.e., large length scales) over desert areas. Similar to the result from Van der Ent and Savenije (2011) we find length scales in the order of 1000 km. We added the figure to our appendix at the end of our revised manuscript and discuss these in lines 378-390.

Although length scale is an important metric, we believe that it is more difficult to apply it to, for instance, the impact of land use change on precipitation locally. For this, one needs to determine the amount of rain that recycles locally, and length scale does not quantify this, whereas the local moisture recycling ratio does. Yet, to apply local moisture recycling locally, we believe it is important to better understand the local moisture recycling ratio first, and with our study we add to its understanding. We added discussion on the differences between the local moisture recycling ratio and this length scale (lines 378-390) to allow readers to better assess what metric to use when addressing research questions related to moisture recycling as we do see value in using the length scale for research questions related to non-local effects.

Concerning the comment on the impact of wind direction, we fully agree that the dominant wind direction affects the value of LMR. We can imagine that especially on higher latitudes, where variation between grid cell shape and size is relatively large and the difference between the zonal and meridional length of the grid cell is large, wind direction may have a strong impact on local recycling. We clarified the possible impact of dominant wind direction in our discussion to create awareness of this effect amongst the readers (lines 366-368).

Finally, dr. Van der Ent states that the analysis in which we study correlations between the local recycling and different variables is fundamentally skewed due to the scale and shape dependency, which some of the other variables don't have. We agree that some of the variables are scale and shape dependent (e.g., CAPE and LMR) and other variables are not (e.g., precipitation and evaporation). This difference between variables might result in a bias in the output of the Spearman rank correlation analysis (see lines 317-319). Therefore, there is no perfect method to make a comparison between recycling ratios in different regions to study the correlation between the local moisture recycling ratio and other variables. However, to address research questions related to quantifying local hydrology, the local moisture recycling ratio is useful and currently no other metric is available to quantify this globally. As comparison of the recycling ratio among different grid cells is difficult to physically interpret (Van der Ent and Savenije, 2011), we need to gain a better understanding of the local moisture recycling ratio. We build this understanding by identifying some

of its drivers. However, based on this comment we conducted another analysis in which we classify the data based on latitude and calculate Spearman correlation coefficients for the data in these different classes. The grid cell sizes within each class will then be more comparable to minimize the skewness of the analysis. The results of this analysis are presented in the results section of our manuscript and in Table 1. Furthermore, because mainly the higher latitudes are skewed, we excluded Antarctica from our analysis.

[Figure]

Figure R1: The local moisture recycling ratio scaled to a grid cell size of 50 km x 50 km. The plot shows the average of 2008-2017. We divided the original local moisture recycling ratio by the area of the grid cell and multiplied it with 2500 km$^2$.

[Figure]

Figure R2: Evaporation recycling length scale as defined by Van der Ent and Savenije (2011) for each grid cell of 0.5x0.5 degrees. The plot shows the average of 2008-2017.

Comment round 2 from dr. Ruud van der Ent

In our response to the first comment by Van der Ent we stated: "Second, this resolution approaches a length scale where local processes have relevant contributions. Feedbacks between the land surface and atmosphere at this spatial scale are relevant for regreening projects. So, we decided to call our metric local recycling as we see potential to use it to study local feedbacks." Van der Ent responded as follows:

I partly agree, but I think it would be good to realize that, as far as I know, no study every looked at the scale at which moisture recycling is the dominant process. But at least we know that at a certain small scale micrometeorological processes are dominant and dry soils or deforested land may sometimes even generate more rainfall triggering (e.g., Knox et al., 2011; Taylor et al., 2012). Whether the local evaporation contribution is 'relevant' at the scale you're studying is, therefore, not beyond doubt. Understandably, identifying that exact scale may be beyond the objectives of this research, but it would be good to acknowledge.

We refer to this point in our discussion in lines 367-376. In this part we use the literature as suggested by dr. Van der Ent in this comment.

Furthermore, in our response to the first comment by Van der Ent we stated: "Additionally, the previous work was conducted using output from a different moisture tracking model (i.e., WAM2-layers), which assumes complete mixing of evaporated moisture within two atmospheric layers. For short time scales complete vertical mixing might not be realistic. The UTrack model distributes the evaporated moisture along the vertical moisture profile and therefore, might be more suitable for analyses on a smaller scale." Van der Ent responded as follows:

Actually, in Van der Ent and Savenije (2011) complete vertical mixing was assumed. Results with the 2-layer model can be found in Van der Ent (2014).

We thank dr. Van der Ent for correcting this mistake. We implemented it corrected this.

Yet, I think you cannot be sure that UTrack is better without performing a similar analysis as we did in Van der Ent et al., (2013), where WAM2layers performed pretty well, and where it was also identified that the way evaporated moisture is distributed over the vertical profile (low or well-mixed) makes a large difference for Lagrangian tracking results. If you use instantaneous vertical well-mixing after evaporation in UTrack it might be fine for large scale tracking, but could very well be unrealistic at the local scale you're studying. Again, probably beyond the scope of your study to evaluate in detail, but it would be good to elaborate on this for your own model and acknowledge any relevant assumptions made.

We agree that we cannot be sure whether UTrack or WAM2-layers performs better in the accounting of rainfall locally. In the revised version of our manuscript, we added a comparison of moisture recycling locally (at a scale of 1.5 degrees) to study the model dependence of local moisture recycling. In our discussion we focus on differences between the two models that might impact the amount of moisture that rains out locally (lines 392-408) without making assumptions on the accuracy of each model.

Additionally, in our response to dr. Van der Ent we stated: "we scaled the local moisture recycling ratio of each grid cell to an area of 50 km x 50 km (see Figure R1)". The response of dr. Van der Ent:

Nice, please describe how you scaled this.

For each grid cell, we normalized the recycling ratio using its grid cell area. Then we multiplied the normalized recycling ratio with the area of 50*50 km (2500 km). We included this information in the caption of the appendix figure (Figure A9).

In our response to the comment by dr. Van der Ent we stated: "Therefore, we need to gain a better understanding of the local moisture recycling ratio. We build this understanding by identifying some of its drivers." Dr. Van der Ent replied to this as follows:

Your explanation above is acceptable, however, it would be good to realize that when you look at the moisture recycling process (Dominguez et al., 2006; equation (20)), the length scale of precipitation recycling emerges from that equation as:

( atmospheric storage * wind speed ) / evaporation

(van der Ent and Savenije, 2011)

thus the length scale of evaporation recycling (here, length scale) is:

( atmospheric storage * wind speed ) / precipitation

Thus if precipitation is high and atmospheric storage and wind speed are low, the local evaporation contribution is relatively high resulting in a low length scale and high local recycling ratio.

When doing your analysis of "drivers", you're thus actually answering the question: what triggers high precipitation compared to low atmospheric storage and little wind speed. Perhaps these thoughts are helpful for your analysis and evaluation of "drivers"

We are thankful for this comment. We used this to improve the formulation of the hypotheses (lines 147-166).

Finally, dr. Van der Ent commented on the following "However, based on this comment we conducted another analysis in which we classify the data based on latitude and calculate Spearman correlation coefficients for the data in these different classes. The grid cell sizes within each class will then be more comparable to minimize the skewness of the analysis." His response was:

That makes sense. Alternatively, you could combine 2 or more grid cells into 1 more square-shaped grid cell from a certain latitude onward. Just a thought.

Besides comments on our response to his first comments, dr. Van der Ent also had a new comment. He mentioned that the symbol P in equations 1-3 might result in confusion.

We agree that using P to indicate the fraction of evaporated water from grid cell $i,j$ that rains out in the source grid cell and its 8 or 25 neighboring grid cells can be confusing. To make it clear, we included a subscript ($P_E$) in all 3 equations to indicate this precipitation originates from the source evaporation.

We thank dr. Van der Ent for this contribution. However, we decided to keep the classification of the data for our Spearman correlation analysis to maintain one definition for local moisture recycling (i.e., recycling of moisture within its source grid cell and 8 neighbouring grid cells).

References provided by dr. Van der Ent:

Dominguez, F., Kumar, P., Liang, X. Z., and Ting, M.: Impact of atmospheric moisture storage on precipitation recycling, Journal of Climate, 19, 1513–1530, 2006.

van der Ent, R. J., Tuinenburg, O. A., Knoche, H. R., Kunstmann, H., and Savenije, H. H. G.: Should we use a simple or complex model for moisture recycling and atmospheric moisture tracking?, Hydrology and Earth System Sciences, 17, 4869–4884, https://doi.org/10.5194/hess-17-4869-2013, 2013.

van der Ent, R. J.: A new view on the hydrological cycle over continents, Delft University of Technology, 96 pp., https://doi.org/10.4233/uuid:0ab824ee-6956-4cc3-b530-3245ab4f32be, 2014.

Knox, R., Bisht, G., Wang, J., and Bras, R.: Precipitation Variability over the Forest-to-Nonforest Transition in Southwestern Amazonia, Journal of Climate, 24, 2368–2377, https://doi.org/10.1175/2010JCLI3815.1, 2011.

Taylor, C. M., de Jeu, R. A. M., Guichard, F., Harris, P. P., and Dorigo, W. A.: Afternoon rain more likely over drier soils, Nature, 3–6, https://doi.org/10.1038/nature11377, 2012.

**Review dr. Patrick Keys**

Thank you for taking the time to review our manuscript 'Local moisture recycling across the globe'. We very much appreciate your feedback as we believe they will help us to improve our manuscript. Below we will shortly respond to each of the general comments you wrote in your review. We will address the specific comments and the general comments in more detail when we get the opportunity to revise our manuscript.

**1. "Drivers": I had some trouble with the word "drivers" being used to describe the role of e.g., CAPE, in LMR. While it's possible that some of the correlated variables could be causally related to LMR, there was no causal analysis completed (or a mechanistic explanation) as far as I could tell. Given that, it seems perfectly reasonable to say correlated phenomena (or similar) with regard to the variables that seemed to have convincing correlations with LMR, e.g., wetness or elevation.**

We agree on the point that a correlation between LMR and any other variable does not imply causality in our study and therefore, we understand that "driver" would not be the best word to use in our manuscript. We thank you for this contribution; we now call them 'factors' instead of 'drivers'. Furthermore, we highlighted in our methods section (lines 169-170) that a correlation does not imply causality in our study.

In addition, we better implemented our hypothesis into our manuscript. We explain which processes we expect LMR to be part of using previous literature in lines 147-166 This supports our choice of variables that we included in our study. For each variable in our study, we also highlight whether we expect a direct relation between this variable and local moisture recycling or whether this variable is considered a proxy (lines 147-166).

**2. LMR Definition: This is purely a suggestion, but I would strongly encourage the authors to frame the LMR idea conceptually first (which I think would force the authors to argue more clearly for the novelty of the idea), and then provide the specific way that they define it in the article (e.g., "LMR is __________, which we define here as _______"). The reason being that LMR could be a conceptually useful idea on its own, but others may make entirely different quantifications, or indeed develop a more robust geophysical definition later on. By distinguishing your conceptual contribution from the physical definition, you may give the concept more scholarly and applied longevity. Again, just a suggestion.**

We very much appreciate this suggestion from dr. Keys. We believe by framing LMR conceptually, our proposal to study local hydrological impacts of land cover changes becomes clearer. This allows future studies to explore the concept of local moisture recycling using different definitions. It also helps us to better highlight the novelty of our approach and we agree that by framing it as suggested by dr. keys we better clarify the importance of moisture recycling locally regarding land cover change. Many thanks for this suggestion we implemented it in the introduction. In lines 61-66 we highlight the concept of local recycling and in lines 86-91 we describe how we define local moisture recycling.

**3. Expand the Introduction: Given the intended scope of this work, the introduction should be expanded to include discussion of past moisture recycling analyses that relate to the LMR idea. There are quite a lot of moisture recycling studies that examine: local recycling (though perhaps not at the global scale), the scale dependence of moisture recycling, and the role of different types of vegetation in the scale dependence of moisture recycling (e.g., van der Ent et al., 2014).**

To better highlight the novelty and aim we expanded the introduction and included more literature on studies related to moisture recycling (lines 26-38). We discuss the length scale of evaporation recycling in our introduction (lines 51-59). We use this to clarify that the length scale is different from the local moisture recycling ratio and show the added value of the local moisture recycling to the scientific community (lines 57-59). We discuss this in more detail in our response to comment 2 from dr. Van der Ent which can be found on page 2 of this rebuttal. Here, we already described in more detail how we implemented the length scale in the introduction and discussion of our manuscript. Thank you for providing these examples. Furthermore, in lines (40-46) we elaborate on previous studies.

**4. Discussion of robustness of a 10-yr climatology: The current draft of the paper does not suitably discuss the appropriateness of the time scale associated with the Utrack dataset in the context of climate variation. Given that the authors are making claims about 'average rates', correlating with phenomena to determine their relationship (potentially causative) of LMR, etc., it is necessary to provide both a justification and a discussion (including of limitations) vis a vis a 10-year climatology. Given that there are inter-annual (e.g., ENSO, Indian Ocean Dipole) and decadal-scale (e.g., Pacific Decadal Oscillation), modes of climate variability that could be systematically affecting some of these results, it is necessary to explain the role of a 10-yr only analysis. Saying that the data are only available for 10 years is not really sufficient. Acknowledging this temporal limitation is critical also for making sure readers can appropriately interpret the results, which could change with a longer time series. Again — I recognize the authors have mentioned the topic of the length of record albeit briefly, but a more in-depth discussion is needed.**

We agree that the 10-year averaged data is not sufficient for a robust climatological analysis and highlighted this better in our manuscript (see lines 434-445). Our analysis, and the 10 years involved, are used for intra-annual variability (seasonality). However, trends in multi-year climate variation could affect our results. This analysis shows that regions that face drier-than-normal conditions due to weather and climate events are characterized by a lower recycling ratio in that same period and vice versa for regions that face wetter-than-normal conditions.

**5. A 50-km definition: The authors select 50 km as the spatial scale of their LMR definition. A bit more discussion is needed to explain the logic and rationale of why such a blanket definition across the terrestrial surface is appropriate, and not an orographically-, latitudinally-, or biome-dependent definition. I'm not suggesting to change away from 50 km, but enough other research has found that moisture recycling ratios and spatial scales are associated with vegetation type, position on continent relative to prevailing winds, proximity to mountains, etc. that the authors need to support their 50km definition more strongly.**

We opted for our definition of local moisture recycling as we needed a systematic definition of local moisture recycling as we focus on moisture recycling across the globe. We fully agree that factors such as orography, latitude and vegetation affect local moisture recycling. Therefore, the aim of our study is to contribute to the understanding of the relation between local moisture recycling and these different factors. However, we added discussion on the role of landscape characteristics on local moisture recycling and how this relates to the spatial scale of local moisture recycling, see lines

367-376. Furthermore, in lines 121-128 we now also elaborate on the choice for recycling within approximately 50 km.

SPECIFIC COMMENTS

1. **L10  Consider saying "defined here as…"**

We changed this line in our abstract as suggested.

2. **L11  You could consider using the phrasing "a 10-year climatology…"**

We implemented this suggestion in our abstract as well as in the rest of our manuscript.

3. **L21  See General Comment about Expanding the Introduction.**

We extended the introduction by including more information on previous studies on moisture recycling.

4. **L23  The sentence beginning with "However, it is unknown…" seems like it might not truly reflect the state of knowledge about moisture recycling globally, and overclaim the knowledge gap. You could consider situating this more concretely in what is understood (e.g., from the perspective of vegetation type having different length scaling associated with moisture recycling, etc.)**

We excluded this statement in the updated version of our introduction. In the revision, we first describe what is understood and then use this to formulate the knowledge gap. This is done in lines: 48-66.

5. **L38  The authors propose to explore "spatial-temporal variation across the globe" but a 10 year climatology is not enough to support the "temporal variation" component.**

In line 79 we changed "spatio-temporal variation" in variation "over the globe and throughout the year for the 10-year climatology" and futher do not mention "spatial-temporal variation".

6. **L67  What is the origin of these correlated variables? I could not see whether they came from ERA5 or somewhere else. Likewise, given the range in geophysical process scale of these variables, it would be good to comment here (or in the discussion of limitations) on the spatial scale of some of these, and whether some are more or less appropriate at the scales of analysis used in the study.**

The data were obtained from ERA5. We included the origin of this data in our methods in line 143-144. It is true that our Spearman analysis is scale dependent. For example, convection can take place on a smaller spatial scale than 50 km. Our analysis does not explicitly capture processes on a spatial scale smaller than 50 km. Therefore, it is possible that, for example, not all convective activity is properly represented by the data we use. However, we do clearly see that variation in CAPE overlaps with variation in LMR, as for high values of CAPE there are no low values of LMR. We expect this that studies on a smaller scale will provide a more detailed output and perhaps a clearer correlation. Yet, we do not expect there will be no correlation between CAPE and LMR when an analysis with a higher resolution is conducted.  We included this issue in our discussion in lines 319-321.

7. **L67  See General Comment about the word "Drivers."**

Throughout the manuscript we changed the word "drivers" to "factors that influence LMR" or something similar.

8. **L78  It is worth noting (either here, or in the Discussion) that the specific continental configuration (and mountain ranges) relative to prevailing winds and the associated composition matters quite a lot for determining the distribution of moisture recycling ratios. I know that the authors know this already (just looking at the author names and their past publications) so it is a noticeable gap in the logic.**

This should indeed be part of this manuscript. We included this aspect in our discussion in lines 367-371. We support this statement by citing the work from Staal et al. (2018) that shows a decreasing median geographic distance of transpiration with decreasing distance from the Andes mountains and the work done by Tuinenburg et al. (2012) that indicates the northward moisture flow originating from the Ganges basin is blocked by the Himalayas inducing precipitation.

9. **Fig3  This is an interesting figure, but I think that the bottom left panel provides the most insight. I would recommend pulling this figure out on its own, so that it can be seen in a larger capacity, given the density of dots and information presented.**

Based on the new spearman rank correlation for more variables, the selection of variables that we show in this figure has become different. We decided to show precipitation, evaporation, CAPE and net surface solar radiation in Fig. 4 (originally Fig. 3) and dedicate a new figure to the scatter plot of LMR and latitude (Fig. 5) as this plot includes most information. The bottom left is the plot of CAPE. However, the bottom right plot (latitude) includes a black line indicating the zonal average of LMR and colors indicating the orography of each grid cell. We expect that Patrick Keys referred to the latitude plot as this includes more information.

10. **L135  Again, drivers is probably not quite right here, since there is no sufficiently causal mechanistic explanation of the link between e.g., CAPE and LMR.**

We agree and changed it into factors underlying LMR

11. **L141  This seems like a perfect opportunity to situate the findings in the sweep of convective storm literature. I suspect that a convective storm meteorologist might not find the statement very surprising "our results suggest a positive relation between convection and LMR." That doesn't mean it shouldn't be said — just that some references aligning this statement with the corresponding literature seem prudent.**

In lines 268-269 we mention the work by Eltahir (1998), Eltahir & Pal (1996) and Williams & Renno (1993) about convective storms and their relation to soil moisture, rainfall and CAPE.

12. **L149  If I'm looking at the right Miyamoto 2013 reference, the argument appears to be that the number of convective features increases dramatically with resolution. That being said, the statement that "convection is a local scale process (i.e., spatial scale of 100 km)" might want to be adjusted.**

We agree with this comment. We changed the phrase "spatial scale of 100 km" to "spatial scale of less than 100 km".

13. **L145  The entire paragraph needs better referencing, since the authors are making numerous claims regarding convective storms, and how the LMR analysis relates to that field. Greater referencing would also give me (the reader) more confidence that these claims are supported by the broader field.**

We included references on precipitation sources for convective precipitation. First, we refer to Eltahir (1998) when we describe why we expect convective precipitation to correlate to LMR. We

refer to Liberato (2012) and Jana (2018) as examples when we explain why convective precipitation can also originate from non-local regions (lines 279-287).

**14. L155   The global biome discussion is very interesting, but in its current presentation it both (a) reads as results, and (b) needs a supporting figure in the main text.**

We are glad that dr. Patrick Keys finds the biome discussion interesting. We agree that it can be read as results in our original manuscript. For our revision, we conducted the Spearman analysis for different latitude classes. These latitudes overlap with biomes. Therefore, this discussion supports the main findings and gives us some more insight into the relevant processes, so we believe this should be part of the discussion. We divided the discussion into separate subsections to increase its readability. We put this paragraph in the subsection 'regional patterns of LMR' (lines 324-335).

**15. L168   These findings are very interesting, and are well discussed. I would use the density of supporting references and citations here as an example of what is necessary in the "convection" section above.**

We are thankful for this positive comment and we used the advice to rewrite the convection section in our discussion as described in specific comment 13 (see above and lines 279-287).

**16. L203   The authors rely heavily (though not entirely) on Salmon et al 2011. Given the range of claims being discussed here, it seems prudent to include a few more references (than relying on Salmon et al. three times in the same paragraph.**

We included three other references in this paragraph (lines 459-474) (Salmon et al. 2011; Molden 2007; Costa et al. 2019; Döll & Siebert 2002).

**17. L221   I encourage the authors to cite the work by Kirsten Findell in this paragraph (see Refs below), who provides a global analysis which blends empirical analysis and theory to explore how continental moisture recycling may change over the coming century.**

We are thankful for this reference. We included it in this paragraph in lines 482-484.

**18. L235   The sentence should be restructured for clarity.**

We split the sentence in two sentences to increase clarity. It now reads: "We expect that the novel concept of LMR can be helpful in various ways. Specifically, we expect that the concept of LMR can be used to study how changes in evaporation, due to for example afforestation, affect the local water cycle beyond merely a loss of moisture."

**19. L242   It might be interesting to be able to state the standard deviation associated with the 1.6% number that is quoted throughout the paper.**

The standard deviation is 1%. We added this in our manuscript everywhere (abstract, results, and conclusion) where we mention the 1.6%.

**Review Anonymous reviewer**

Thank you for your time and for reviewing our manuscript: 'Local moisture recycling across the globe'. We appreciate your feedback, which is very helpful to improve our manuscript. Below we respond to each major point of feedback separately to discuss how we will implement them. The minor comments will be addressed when we get the opportunity to revise our manuscript.

**1. Novelty**

The manuscript repeatedly claims that local moisture recycling ratios are calculated "for the first time" (l. 9) and that "it is unknown which fraction of moisture recycles within its source location, and how this recycling varies across the globe" (l. 22-23). However, this is not the first study to do exactly this: Van der Ent et al. (2010) and Van der Ent & Savenjie (2011) already featured such local evaporation recycling ratios and calculated them globally. Furthermore, 'evaporationsheds' (see e.g. Van der Ent & Savenije, 2013) contain the exact same information and papers and data sets have been published on this, see e.g. Link et al. (2020).

Unfortunately, I also cannot consider the approach, or the objective referred to in the discussion novel: the perspective on understanding the potential influence of land cover, and land- and water management practices via moisture recycling is not new either. Keys et al. (2016), for example, describe this in the context of 'ecosystem services' or 'water security' (Keys et al. 2020) - to name just a few examples. And this is also the subject of all 'green water' studies (e.g., te Wierik et al., 2021; te Wierik et al. 2020).

We thank the reviewer for explaining why they believe our manuscript is not as novel as stated in the manuscript. Considering that the comment posted by Dr. Ruud van der Ent, the review posted by Dr. Patrick Keys and this review all include this point about novelty, we see the importance of improving on this point. Therefore, we better acknowledge all relevant previous studies, for example the studies mentioned by the reviewer in the introduction (lines 23-46). Thus, we were better capable of highlighting the differences between our work and previous work as we do believe there are important new steps being made in our paper. Namely, first, we study the effect of the spatial scale of local moisture recycling of our definition of local, i.e., the area within which the moisture recycles. Second, we assess potential drivers of local moisture recycling. Furthermore, the datasets that were used to calculate local recycling in previous studies differ from the dataset used in our study. To obtain these datasets, different models with different input data were used. We mention a different model in our introduction (lines 51-53) and come back to it in our discussion in more detail (392-408). Furthermore, we highlight how our work compares but also deviates to the work done related to 'ecosystem services', 'water security' and 'green water' studies (lines 40-46), to which the reviewer refers. Those studies have a focus on source-sink relations in which the sink, apart from the source region, includes also remote locations. In contrast, our work aims to quantify and better understand local recycling as land use change can also alter the water balance locally. For example, regreening can cause local drying.  This illustrates the relevance of studying the impact of land cover changes on the local water cycle. Even though Van der Ent and Savenije (2011) calculated a similar type of recycling, the link with preventing local drying has not been made yet, which we believe could be highly valuable (lines 61-66). In addition, the spatial scale of 0.5 degrees better allows studying local impacts than the scale of 1.5 degrees. We are thankful for this comment as it helps us to specify the novelty of our manuscript better.

2. Moisture recycling drivers

I do, however, like the idea of looking at the drivers of moisture recycling; but the current analysis of the drivers is rather simple. In particular, I am a bit hesitant about the variables used to unravel the drivers of LMR, and the methodology used to do so. First of all, while I understand that there is a latitudinal dependence of moisture recycling, I wonder if 'latitude' is the real driver here. Shouldn't it rather be wind, incoming solar radiation and maybe even the underlying area of a grid cell (that differs with latitude)? Similarly, is it fair to use 'evaporation' and 'precipitation' as drivers of LMR? Isn't LMR defined based on these two fluxes? Of course, there is a dependency on both fluxes then... Second, calculating (globally averaged?) Spearman correlations to unravel drivers of

**LMR is a cheap way of doing this. LMR and any variable in Tab. 1 may be correlated through a third variable that represents the 'true' driver. Or in other words: a correlation does not imply causality.**

We are happy that the reviewer likes the aim to understand the drivers of local moisture recycling but also understand why the reviewer likes to see more regional (latitudinal) tests that can potentially provide more understanding. In our study we aimed to identify non-linear relations between two variables as most processes cannot be properly described using linear relations and, therefore, we used Spearman rank correlations. Of course, we agree that correlation does not imply causality and to clarify this we now specifically mention this in the methods of our manuscript (lines 169-170). Moreover, we used more literature in the methods section to discuss our hypothesis from a mechanistic point of view (see also response to comment number 1 by reviewer Keys and lines 147-166). In addition, added more references in our discussion where we try to physically explain the correlations (lines 259-296). Furthermore, we agree that latitude is not the actual driver of moisture recycling, but that other variables that correlate with latitude would drive local recycling. As such, we included latitude as a proxy for a combination of processes that have a strong latitudinal pattern. This was not properly described in our manuscript and therefore, we clarified this in our revision (e.g., lines 268, 281, 285-286). We also speak of 'factors that correlate to LMR' instead of 'drivers' to be more clear that the found correlations do not imply causality.

Furthermore, we added more factors (formerly drivers) to study the correlation between local moisture recycling and other potential factors of local moisture recycling, such as solar radiation and wind, as suggested by the reviewer (lines 162-165 and Table 1). We did not move to multiple regression models, as suggested by the reviewer. Multiple regression models assume linear relations and because of the non-linearity of the processes involved we decided to not use multiple regression models. We kept the Spearman rank correlation test per driver. In addition, we split the data in classes based on latitude (lines 170-173) to account for the decreasing grid cell size with increasing latitude. These classes are defined as follows: class 1: 0°-15°, class2: 15°-30°, class 3: 30°-45°, class 4: 45°-60°, class 5: 60°-75°, and class 6: 75°-90°. In each class, the grid cells of the northern and southern hemispheres are included. However, classes 5 and 6 only contain grid cells of the Northern Hemisphere as on the Southern Hemisphere the only land surface between 60° and 90° is Antarctica, which is excluded from our analysis. We believe that adding more variables to our analysis has improved our understanding of local moisture recycling and we are thankful for this comment.

**3. Issues of scale**

**The definition of what is considered 'local' is rather random. The authors claim that the LMR is based on approx. 50km around the source; however, they also illustrate different definitions of this scale parameter, i.e. 1 grid cell, 9 grid cells and 25 grid cells. The argument for chosing 9 grid cells is rather vague: "To keep the spatial scale as small as possible but to still have a spatial pattern that we can explain physically" (l. 88-89). Could the authors explain why other patterns cannot be explained physically? Is there some lower limit to what the forcing and/or the model can represent? If so, could this limit be determined in a reasonable manner?**

We thank the reviewer for pointing out the unclarity concerning the spatial scale of local moisture recycling. We agree that the definition is partly arbitrary. Concerning the decision to use recycling over 9 grid cells, local recycling within one grid cell results in exceptionally low values over mountain peaks, yet not over all elevated terrain and relatively high values over the ocean. This pattern is inconsistent with the result found for recycling within 9 and 25 grid cells. The patterns for recycling over 9 and 25 grid cells can be explained as high values over mountains can result from convection because of orographic lift and relatively low values over ocean can be explained by the large

atmospheric moisture transport due to strong winds. We omitted the sentence in which we state we cannot "physically" explain the pattern of recycling within one grid cell. We now only state that the pattern does not overlap with recycling over 9 ad 25 grid cells. However, this does not support our decision sufficiently and we thank the reviewer for pointing this out.

The main reason not to use $r_1$ in our study is that for recycling within one grid cell, the moisture flows to one of the surrounding grid cells can still have a length smaller than 50 km, i.e., moisture can evaporate on the border of a grid cell and rain out in the center of the grid cell located next to it. These flows also describe local recycling and should therefore be included in LMR. These flows are included in $r_9$. This explanation was not described in the original version of the manuscript. We included it in the revised version in lines (121-123).

Concerning the last question the reviewer includes in this comment, our results show that at this resolution $r_9$ is the lower limit of LMR. However, the UTrack model can provide output at a spatial scale of 0.25°. Therefore, the lower limit is $r_9$ at a spatial resolution of 0.25°. In the future, when the input data of the UTrack model is available at a higher resolution, the lower limit might be lower than it currently is.

**Some suggestions**

**To make this a novel and interesting contribution in the field of moisture recycling, a bit more effort may be needed. The authors could, for example, compare their evaporation recycling ratios to the ones from Link et al. - I assume that much more could be learned from the difference of these data sets. Alternatively, the 'true' drivers of moisture recycling could be assessed, using a more sophisticated method to do so. Or the issue of scale and what can be considered local, given the spatio-temporal resolution of the forcing, could be put into focus... these are, however, just some suggestions that I could envision and that would make this paper novel and interesting to me. The authors do not need to follow those.**

We agree with these suggestions. As described before, we conducted more analyses by studying the relations between local moisture recycling and surface net solar radiation, wind speed, boundary layer height and total cloud cover. In addition, we classify our data based on and conduct a spearman rank correlation analysis for each class separately (lines 147-173). Furthermore, we compare our results to the dataset by Link et al. (2020) (lines 130-135). For this we calculated LMR on a spatial scale of 1.5 degrees to match the dataset from Link et al (2020. This comparison gives some insight into the model dependence of LMR. We find that the spatial pattern of recycling within one grid cell of 1.5° is similar for the different models. However, its magnitude is strongly model dependent (lines 206-212).

Minor points

1. l. 52-53: "Parcels are tracked for up to 30 days or up to the point at which only 1% of their original moisture is still present. " - can this be longer than 30 days?

It is possible to track parcels longer than 30 days. However, we obtained the data from a readily available dataset for which moisture tracking was done for 30 days. Tuinenburg and Staal (2020) describe that the accuracy of tracking decreases with tracking time and that many tracking models only track moisture for 10 days because of this reason. However, within a period of 10 days not all moisture in the parcel might have rained out yet (Sodemann, 2020). Therefore, Tuinenburg et al. (2020) decided to track parcels up to 30 days. It is possible that for a few parcels not all moisture has rained out of the parcel after 30 days (Sodemann, 2020). Tuinenburg

and Staal (2020) considered that (1) the tracking time affects the accuracy of the output and (2) not all moisture has left the parcel after 10 days and to account for both issues they used a tracking time of 30 days. We added lines 96-97 to clarify the decision of 30 days.

2. **Eq. 1-3: this refers to different areas across the globe; where do the 50km from the abstract come in?**

These equations show the three different definitions of the local moisture recycling ratio of a grid of 0.5 degrees. We define LMR as the recycling of evaporated moisture in its source grid cell and 8 surrounding grid cells. The distance from the center of the source grid cell and its surrounding grid cells describes the typical length of the moisture flow. We calculated the length of the average moisture flow across the globe by calculating the average zonal length, meridional length and diagonal length of all terrestrial grid cells. The average of the zonal length equals 39.28 km, the average of the meridional length equals 55.60 km, the average of the diagonal length equals 55.55 km. The total average equals 50.14 km (st.dev. = 15.5 km). Therefore, the average moisture flow length is approximately 50 km. We added this explanation in the methods (lines 126-128).

3. **Uncertainty of UTrack is not assessed at all; at least assumptions in the model should be summarised in the Methods section as well.**

It is difficult to assess the uncertainty of the model as there are no global observations we can compare our output to. However, in our revision we included a comparison with the local recycling calculated using the output from WAM2-Layers by Link et al. (2020). This comparison shows there is little uncertainty in the spatial patterns. However, there is a large uncertainty in the magnitude of recycling within 1.5 degrees grid cells (lines 206-212). We discuss the differences between the models in more detail in the discussion (lines 392-408). Furthermore, Tuinenburg and Staal (2020) conducted a sensitivity analysis to study the influence of different model settings on the atmospheric moisture connections. In this sensitivity study the output was compared with a baseline model that incorporates as much detail as possible. This baseline model is a three-dimensional Lagrangian model with interpolated wind speeds and directions. This model releases 10 000 parcels for each mm of evaporation. Based on this sensitivity analysis, the most efficient (accurate and time efficient) tracking method was developed which was used by Tuinenburg et al. (2020). We refer to this work in lines 99-100. Furthermore, Staal et al. (2020) studied the influence of the time step between the random vertical mixing. They found that their output was generally weakly influenced by it. However, they also found that higher vertical mixing causes forest evaporated moisture to rain down more locally.

4. **Fig. 1: it should probably be "grid cell" and not "grids" in the subtitles**

We changed grids to grid cells in the subtitles of the plots in Fig. 1. This figure is now part of the additional information as Fig. A2.

5. **l. 85-85: "These results seem to indicate that the tracking method we use is not sufficient to define recycling within one grid cell."; maybe it's not the tracking method but the (temporal) resolution of the forcing that is used, or the number of parcels tracked?**

It could indeed be that the numerical resolution of our input data is not sufficient and causes this problem. Taking into account the feedback of all reviewers and all community commenters we made some changes in the paragraph in which we compare the different local recycling definitions. We omit the statement that the tracking method might not be sufficient and support our decision to use recycling over 9 grid cells as our definition of LMR (lines 115-135).

6.  **l. 86-87: "Finally, scaling recycling to the number of grid cells, we find r9 and r25 do not relate linearly." Could you elaborate how you scaled this? P is not uniformely distributed across the 9 or 25 grid cells considered here, so I would not expect that there is a linear relationship?**

Indeed, we agree with this point. As p is not uniformly distributed, we would not expect a linear relation between 9 and 25 grid cells. For most footprints, further way from the source, the amount of precipitation originating from this source becomes smaller, resulting in a non-linear relation. We moved the paragraph that included this sentence to the methods (lines 115-135), and we removed this line from this paragraph as it does not provide new insights.

7.  **A suggestion: a uniform color scheme for Figs. 1-2 would be helpful**

The same color scheme was used for Figs. 1-2. The color bar of the middle plot in Fig. 1 resembles the color bar of Fig. 2. Note that in Fig. 2 LMR is only presented over the land surface and the oceans and sea have a gray color. After revising our manuscript, Fig. 1 is now part of the appendix as Fig. A2 and the original Fig. 2 is now Fig. 1.

8.  **l. 111-114: "Both convective and large-scale precipitation correlate with LMR (Table 1), however neither the fraction of convective precipitation nor the fraction of large-scale precipitation correlates with LMR (Table 1). Furthermore, evaporation correlates positively with LMR (Table 1, Fig. 3) indicating that the strong relation between P and LMR is not the only factor that causes a correlation between wetness and LMR." - does it make sense to correlate LMR with P? And as LMR is based on E and P, it needs to be correlated to E as well, right? E could also correlate with LMR because of P... there are so many dependencies here that it is difficult to unravel the real drivers.**

LMR is indeed dependent on many different variables, and also these variables can be dependent on one another. However, we do not agree with that E necessarily correlates with LMR, as for locations with high E and low P, LMR is expected to be small. We agree that it is not possible to find the real drivers. As pointed out by dr. Patrick Keys, it would be better to use different terminology than 'drivers'. Throughout the revision we use 'underlying factors' instead of 'drivers'. Despite the difficulty of identifying the real drivers, understanding which variables correlate to LMR might help us to predict local precipitation under changing conditions (e.g., land use change and climate change). However, to be able to make predictions, we also need information on how evaporation and precipitation correlate with LMR and therefore, we included them in our study. We included this issue in our discussion in lines 319-321.

9.  **The relation between LMR and convection is not surprising; however, what would be novel was if large-scale and convective precipitation were tracked separately...**

We agree with the reviewer that this would be novel, but this would be a study in itself. To do so, the state-of-the-art moisture tracking models would need some alternations to enable this tracking. It would be interesting and possible to calculate the amount of convective precipitation and large-scale precipitation for each grid cell while tracking. However, in reanalysis data, the type of precipitation is calculated using a parametrization, which causes large uncertainty. Therefore, tracking convective precipitation and large-scale precipitation separately might cause significant biases.

10. **l. 146ff: Are the correlations, especially with convective precipitation and large scale precipitation, subject to spatial and temporal scales?**

Some correlations are indeed subject to spatial and/or temporal scales. For example, convection is stronger in summer than in winter for many grid cells (lines 313-317). On the other hand, correlations with wind are dependent on the spatial scale as for a smaller grid cell length moisture is transported out of the cell quicker than for a larger grid cell, assuming equal wind speed (lines 366-367). Yet, most importantly is that LMR is subject to spatial and temporal scale. To account for differences in grid cell size we classified our data based on latitude, and for each class we conducted the Spearman analysis. This classification divides grid cells in groups based on length scale and prevents our analysis to be skewed due to different grid cell sizes. We discuss this issue in our discussion in lines: 361-367.

**11. l. 155-174: discussion on biomes and deforestation a bit misplaced; not motivated in the introduction at all**

We agree that the discussion on biomes was a bit misplaced. In our revision we link our Spearman analysis for the different latitude classes to the discussion on biomes (lines 324-335). We believe it provides some additional information on the differences between the different latitude classes. To better implement deforestation into our manuscript we included it in two paragraphs in the introduction. See line 38 and line 64.

**12. l. 179-181: well-mixed assumption is often hidden in many tracking studies; as far as I understand this is also the case for UTrack - and a recent study illustrated the impact using another Lagrangian model (Keune et al., 2022)**

In Lagrangian and Eulerian moisture tracking models the vertical mixing is parameterized differently. However, some models do not include a completely vertically mixed atmosphere but track moisture on different vertical atmospheric levels. For example, WAM2-Layers tracks moisture over two vertical levels to account for vertical shear. These two layers have a different specific humidity and therefore, moisture is not well-mixed in the vertical direction. UTrack tracks moisture over 25 vertical levels with each level having its own specific humidity and therefore the lower layers contain more moisture than the higher vertical layers. Over these layers the evaporated moisture is divided partially randomly so indeed there is still mixing over the total height of the column. This is discussed in lines 418-429. The study that the reviewer mentions (i.e., Keune et al., 2022) uses FLEXPART that accounts for mixing using a different method. In this study, the input data has a time step of 3 hours but the timescales of the trajectories are smaller to increase the interaction between different wind components. This improves the representation of turbulence. In addition, this study takes into account sub-grid terrain effects, improving mixing. We included this study in our discussion as an example of an alternative method to include mixing in a tracking model (lines 429-431).

**13. l. 185ff: this should really be described in the methods, in my opinion**

We agree and, therefore, we removed this sentence from the discussion. It is now described in our methods.

**14. l. 207: relation to agricultural water management remains unclear to me**

We agree that the relation to water management and the first sentences of this paragraph might have been a bit vague. Therefore, we divided this paragraph in two parts. The first part, in which we discuss the robustness of the spatial pattern is included in a paragraph that describes differences in local recycling for UTrack and WAM2-layers (lines 392-395). The second part in which we discuss the relation of LMR to agricultural water management is found in different paragraph. We included this

part to give an indication of how LMR can be interpreted and applied. This part can be found in lines 459-474. Finally, we included the relation of LMR and agricultural water management in the introduction of our manuscript (lines 43-45 and 63-65).

**15 l. 234f: while I understand that you aim to use the LMR as a proxy for regions, in which land and water management may help foster moisture recycling, I don't think this scales at all. To assess the potential of the LMR as a proxy, it would be useful to know if, e.g. an increase in local E by say 10% also leads to an increase of local P by 10%. As you discuss correctly: there are many more factors that play a role here - not just the average recycling ratio; and I am missing an attempt to look at the 'true' drivers of LMR or at least an analysis that moves towards a better suited proxy to estimate the benefit/loss of water due to land- and water management practices in the conext of moisture recycling...**

It is indeed likely that a 10% change in evaporation due to land cover changes might not result in a 10% increase in precipitation. To find the exact change in precipitation we need to study the impact of land cover changes on the atmospheric moisture connections. However, for this we need input data for UTrack including evaporation, precipitation, wind speed in three dimensions and the total precipitable water content that is derived with an atmospheric model that used a future land cover scenario as input. Currently, this is not available and therefore, we try to identify correlations between LMR and other variables to assess which variables have an important influence on LMR to get an idea of how LMR could change in the future due to land use changes. We agree that this study does not reveal the exact processes that underly LMR, and to what extent. However, this study explores what processes are linked to LMR.  We included this issue in our discussion in lines 319-321.

References:

Link, A., van der Ent, R., Berger, M., Eisner, S., and Finkbeiner, M.: The fate of land evaporation – a global dataset, Earth Syst. Sci. Data, 12, 1897–1912, https://doi.org/10.5194/essd-12-1897-2020, 2020.

Van der Ent, R. J., & Savenije, H. H. G. (2011). Length and time scales of atmospheric moisture recycling. Atmospheric Chemistry and Physics, 11(5), 1853–1863. https://doi.org/10.5194/acp-11-1853-2011

Van der Ent, R. J., Savenije, H. H. G., Schaefli, B., & Steele-Dunne, S. C. (2010). Origin and fate of atmospheric moisture over continents. Water Resources Research, 46(9), W09525. https://doi.org/10.1029/2010WR009127

Keys, P. W., Wang-Erlandsson, L., and Gordon, L. J. (2016), Revealing invisible water: moisture recycling as an ecosystem service, PloS one, 11(3), e0151993.

te Wierik, S. A., Cammeraat, E. L. H., Gupta, J., & Artzy-Randrup, Y. A. (2021), Reviewing the impact of land use and land-use change on moisture recycling and precipitation patterns, Water Resour. Res., 57, e2020WR029234. doi:10.1029/2020WR029234.

te Wierik, S. A., Gupta, J., Cammeraat E. L. H., Artzy-Randrup, Y. A., (2020), The need for green and atmospheric water governance, WIREs Water, 7:e1406, doi:10.1002/wat2.1406.

Tuinenburg, Obbe A., Theeuwen, J. J. E., & Staal, A. (2020). High-resolution global atmospheric moisture connections from evaporation to precipitation. Earth System Science Data S, 12(4), 3177–3188. https://doi.org/10.5194/essd-12-3177-2020

---

## Author Response (AR2)

Dear dr. Alexander Gruber,

Thank you for giving us the opportunity to further revise our manuscript "Local moisture recycling across the globe" for *Hydrology and Earth System Sciences*. We would also like to thank dr. Ruud van der Ent and the anonymous reviewer for reviewing the revised manuscript and for their valuable feedback. We were happy to see only minor changes are requested to improve our manuscript. To improve our manuscript, we made some changes to clarify the newly added analyses, the novelty of our study and rephrased some sentences to improve the text overall. In addition to the changes made to the manuscript following the feedback from the reviewers, we made some additional corrections. First, we included an updated version of Figure 5. Second, we included a copyright statement in the caption of Figure A4.

Below, we discuss all comments of reviewers and describe how we implemented them in our revised manuscript. We do this for each review separately. Our response to each comment is presented in blue.

We hope that our manuscript is now considered suitable for publication in *Hydrology and earth System Sciences*. In case of further questions or requests, please let me know.

On behalf of all authors,

Yours sincerely,

Jolanda Theeuwen

**Review by Dr. Ruud van der Ent**

**The authors did a lot of work improving this paper, perhaps even more than was necessary.**

We thank the reviewer for acknowledging the work we put in our revised manuscript.

**My comments were all addressed appropriately, but I do have new comments related to the new comparison analysis with WAM2layers results**

**Figure 2 should be extended by a panel showing the annual Utrack results**
We included this extra plot in Figure 2.

**In Figure 2 It surprises me a lot that it appears that WAM2layers has higher local recycling. Surprising as I remember from previous papers using Utrack that it had higher continental recycling than WAM2layers. Moreover the differences are not so strong for the evaporation length scale comparing visually your Figure A10 with Van der Ent (2014, Figure 3.4b). If anything Utrack length scales are generally lower, implying higher local recycling. Thus, please double check that you are actually comparing 1.5degree results with 1.5degree results are compare length scales instead.**

We checked whether we are actually comparing 1.5 degrees results with 1.5 degrees results in figure 2 and we can confirm that this is the case. However, Fig. A10 shows the evaporation recycling length scale calculated at a resolution of 0.5 degrees, which might explain the differences in length scale compared to the result of Van der Ent (1.5 degrees resolution).

As the recycling ratios obtained from WAM2-Layers are larger than our results obtained with UTrack, we would expect the length scale to be smaller for the output of WAM2-Layers compared to our results obtained with output from UTrack, as explained by the reviewer. However, we cannot

properly compare the evaporation recycling length scale, as originally, we calculated the evaporation length scale at a resolution of 0.5 degrees. Therefore, for this revision, we calculated the evaporation recycling length scale at a resolution of 1.5 degrees to enable a comparison. The results are presented below.

[Figure]

**Figure R1. Length scale of evaporation recycling calculated from a resolution of 1.5 degrees obtained with output from the UTrack model. We used the data from Tuinenburg et al. (2020)**

[Figure]

**Figure R2. Difference in length scale of evaporation recycling calculated from a resolution of 1.5 degrees obtained with output from UTrack and WAM2-Layers. This plot presents length scales obtained with UTrack minus length scales obtained with WAM2-Layers. We used the data from Link et al. (2020)**

First, comparing the length scale of evaporation recycling calculated from a resolution of 0.5 degrees and 1.5 degrees (Fig. R1), we find the length scales are larger for the latter, suggesting we cannot compare the length scale of recycling obtained at different resolutions. A smaller length scale obtained from recycling within 0.5 degrees, compared to the length scale of 1.5 degrees could be the result of a non-linear relation between precipitation and distance to the source.

Second, comparing the length scale at similar resolutions for the output of the two different models, we find that the length scale is larger for the output of UTrack, which suggest lower recycling ratios (Fig. R2). Additionally, the length scale calculated from a resolution of 1.5 degrees is no longer smaller than the length scale found by Van der Ent and Savenije (2011).

To summarize, in the comment of the reviewer the evaporation recycling length scale was calculated at a different resolution, causing the surprising deviation. Furthermore, the evaporation recycling length scale and recycling ratios of 1.5 degrees complement each other. For UTrack, we find lower recycling ratios and larger length scales than for WAM2-Layers.

**As the local recycling ratios are really small numbers, especially over dry areas, the relative differences as shown in Figure 2 are not necessarily so meaningful to compare.**

We thank the reviewer for pointing this out. In the first revision of the manuscript, we referred to the difference in the recycling ratios produced by UTrack and WAM2-Layers over drylands and deserts, as here, large deviations are found. In our second revision we included a statement that the relative differences are less meaningful (lines 190-191): *"However, over drylands and deserts recycling ratios are relatively small and therefore, the relative difference as presented in Figure 2 has less meaning here."*

**In A5 you are comparing different resolutions isn't it? At a minimum mentioning it would be good to mention, but perhaps just scale this to a common 1.5 degree again.**

In Figure A5 we are making a comparison of recycling within one grid cell of 1.5 degrees obtained from the output from Link et al. (2020) and recycling within one grid cell of 1.5 degrees obtained from the output from Tuinenburg et al. (2020). So, we did scale the latter to a 1.5 degrees resolution. We clarified this in the caption of Figure A5: *"The 10-year climatology (2008-2017) of the recycling within one grid cell of 1.5° calculated with the dataset by Link et al. (2020), i.e., the output from the Eulerian moisture tracking model WAM2-layers (top) and the difference with the The 10-year climatology (2008-2017) of the recycling within one grid cell of 1.5° calculated with the dataset by Tuinenburg et al. (2020)."*

**In Equation (4) 'i=jan' and 'dec' should swap position.**

We thank Ruud van der Ent for pointing this out. We corrected it as suggested.

The code and data availability statements are not following the HESS policy https://www.hydrology-and-earth-system-sciences.net/policies/data_policy.html. Code to a least reproduce the figures should be available or a good reason why not should be provided. Data underlying the manuscript should be your produced data, not your input data, which was already mentioned in the methods section.

We uploaded the local moisture recycling ratio data to the Zenodo archive (10.5281/zenodo.7684640). and uploaded the scripts that can be used to plot these data to GitHub (https://github.com/jtheeu/LocalMoistureRecycling). In the revised code and data availability statements we included links to the data and scripts.

**Review by anonymous reviewer**
* Summary and recommendation

**I thank the authors for answering all comments in detail and acknowledge that many comments have been addressed. In particular, I appreciate the attempt to compare the recycling ratios with another data set (Link et al., 2020) and the discussion thereof. The manuscript structure and the introduction have improved considerably.**

We thank the reviewer or acknowledging the improvement of our manuscript.

**However, honestly speaking, I still miss the novelty and the plausibility of this research, which stems from a few shortcomings:**
**- The concept of "local moisture recycling" (whichever grid cell size and model you use) is not**

novel.
- The analysis of "factors influencing local moisture recycling" remains superficial and does not provide much insights.
- There remain a few logical inadequacies.
- The analysis is based on 10 years of simulations only. I could ignore this, but in combination with the points above, this really limits the novelty and the scientific findings that could be achieved.

Following this reasoning, I am afraid that I suggest another round of major revisions.

I elaborate on all issues (except the latter one) with a bit more detail below.

* Novelty of the concept

I acknowledge that the authors did an effort to improve the description of the novelty of the study and I thank them for clarifying a few (minor) differences. I still think that the novelty is lacking though, for two main reasons:

First of all, because the concept of the LMR is not new and the authors 'just' calculate it over a different scale and with a different model than previous studies. Statements such as
l. 9-12 "For the first time, we calculate the local moisture recycling ratio (LMR) as the fraction of evaporated moisture that precipitates within a distance of 0.5° (typically 50 km) from its source, identify variables that correlate with it over land globally and study its model dependency."
l. 234-235 "but for the first time, we analysed the local moisture recycling ratio (LMR) (of evaporated moisture) across the globe at 0.5° resolution"
are on the edge of what you can call "for the first time", in my opinion. There is no scientific novelty in that other than the fact that you use another resolution.

We removed 'for the first time' from both the abstract and discussion in lines 9-12 and 234-235.

Second, the correlation analysis that 'replaces' the driver of moisture recycling idea does not add much information for me either. In fact, the entire section 3.2 became a bit difficult to read because the authors have to remain superficial with their statements. However, if I do not consider the concept novel, this is the only part that could provide novel insights. At the moment, I struggle seeing any novel insight though.
Throughout the entire paragraph we rephrased sentences to improve the readability (lines 197-222)

 * Drivers / "important factors for local moisture recycling"

The authors decided to stick to the calculation of correlations (instead of using causality measures) and refer to variables that correlate with LMR as "important factors for local moisture recycling" instead of "drivers". That is better, although it also implies a direction through the "for", i.e. that the other variable is important for LMR. However, it could also be the opposite: LMR influences the other variable or LMR is important for the other variable. Example: P is not influencing local moisture recycling, but local moisture recycling is influencing P. Which way is it? I honestly don't think even this simple example is easy to answer, and this may be my problem with the setup of this analysis. The way this analysis is referred to and presented in the manuscript is tricky in several parts because of this. A few examples:
- l. 130 "to identify factors that affect recycling"

- l. 133 "To identify factors that affect LMR, …"
- l. 234-236 "but for the first time, we analysed the local moisture recycling ratio (LMR) (of evaporated moisture) across the globe at 0.5° resolution, and which factors affect it."
- l. 254-255 "As both correlations are similar, this suggests that the type of precipitation does not affect LMR."
- l. 284-285: "We aimed for a general analysis to identify the main factors that influence LMR"
and others. I do understand that expert knowledge is used for some of this, but I really struggle to understand this.

**The manuscript should be carefully checked for such implications. If the authors wanted to assess the direction using correlations, they could include time lags in the correlation analysis.**
We thank the reviewer for pointing out some examples in which we imply that we assume a relationship because of a correlation. We used these examples to rephrase the following sentences:

- Line 130: *"… to identify factors that  relate to recycling to assess what factors might affect recycling."*

- Line 135: *"To identify factors that might affect LMR …"*

- Line 242: *"… and which factors might affect it"*

- Line 244: *"First,  latitude, elevation, and Convective Available Potential Energy (CAPE) seem to be  important factors influencing LMR (Fig. 5)."*

- Line 257: *"…  wetness  seems to be an important factor underlying LMR …"*

- Line 261: *"… this suggests that the type of precipitation  might not affect LMR …"*

- Line 291: *"We aimed for a general analysis to identify the main factors that might influence LMR"*

- Lines 450-451: *"We find a correlation between LMR and  orography, precipitation, wetness, convective available potential energy, and wind  suggesting these variables might affect LMR."*

We also would like to thank the reviewer for the suggested analysis. We will take it into consideration for future projects.

**Logic inadequacies**

**- Spatial resolution: I got a bit confused by the spatial resolution used. First of all, in l. 87-90, the authors say "These moisture connections are a 10-year climatology (2008–2017) of monthly averages and have a spatial resolution of 0.5°. These UTrack-atmospheric-moisture data are derived using a Lagrangian atmospheric moisture tracking model by Tuinenburg & Staal (2020) that tracks evaporated moisture at a spatial scale of 0.25°. Could you clarify why there is this discrepancy? You seem to be using 0.5° after all.**
The moisture tracking model UTrack, tracks moisture that is transported over grid cells that have a spatial resolution of 0.25°. Tuinenburg et al. (2020) published a dataset that was used in this study in which they stored the UTrack output on a spatial resolution of 0.5°. Tuinenburg et al. (2020) did not store this dataset with a resolution of 0.25° due to the large size of the documents. The dataset of 0.5° uses 120GB of memory in total. We specify in our manuscript that the moisture is tracked at a resolution of 0.25°, as the tracking resolution

might affect the output of the model. We clarified this by adding further explanation in line 83: *"and stored at a spatial resolution of 0.5°."*

**- High-resolution local moisture recycling: in their response and in the manuscript, the authors argue that they present "high-resolution" local moisture recycling ratios for the first time. They argue that their recycling ratios are calculated over a considerably higher resolution than the 1.5° from van der Ent and Savenjie (2011), see e.g. l. 73 "Moisture recycling has not been studied before on this high-resolution scale globally." However, all recycling ratios are calculated over the source grid cell and the nine surrounding grid cells of 0.5° (r9), hence there is not much difference. In fact, this is pretty much the same 1.5° area that is considered. I am not saying that the authors cannot do this, but there is some major logical flaw in this argumentation, that, unfortunately, does not make this (technical) aspect novel either.**

As the reviewer describes, we indeed calculate the recycling of evaporated moisture from 1 grid cell of 0.5° over 9 grid cells of 0.5°. However, this is different from the work done by Van der Ent and Savenjie (2011) as in their study moisture evaporates from the total area of 1.5° in which it recycles. In the definition of Van der Ent and Savenjie (2011) moisture flows of 1.5° are included as all moisture flows from one edge of the grid cell towards the opposite edge of the grid cell (1.5° distance) are included. In our study we only include moisture flows up to 0.5° as the source region is smaller than the target region and the source region is located in the centre cell of the target region. To clarify this we made the following changes: First, *"For r9, the distance between the center of the source grid cell and its surrounding grid cells describes the typical length of the local moisture flow, which is approximately 0.5°."* (lines 117-118). Second, *"Compared to $r_1$, this $r_9$ includes all moisture flows with a length scale of typically 50 km (0.5°)."* (line 325).

**\* Minor comments**

**- Calculation of correlations: I searched through the entire manuscript but I couldn't find it. I believe the correlations in Tab. 1 are based on the climatological means in each grid cell, i.e. the authors use one P value and one LMR value for each grid cell and then calculate a correlation of all P and all LMR values in a specific cluster. Is that correct? Or do you consider the temporal dynamics in each grid cell?**

This is indeed correct. We clarified this in line 139: *"For all variables we calculated the climatological mean for the years 2008-2017"*. We thank the reviewer for pointing this out as it helped to improve the clarity of our manuscript.